# NuwaDynamics: Discovering and Updating in Causal Spatio-Temporal Modeling

**Kun Wang**[2,†], **Hao Wu**[4,†], **Yifan Duan**[3], **Guibin Zhang**[6], **Kai Wang**[8],
**Xiaojiang Peng**[7], **Yu Zheng**[9], **Yuxuan Liang**[5*], **Yang Wang**[1,2,3,4*]

[1] Key Laboratory of Precision and Intelligent Chemistry, University of Science and Technology
of China (USTC) [2]Suzhou Institute for Advanced Research, USTC
[3]School of Software Engineering, USTC [4] School of Computer Science, USTC
[5] Hong Kong University of Science and Technology (Guangzhou) [6] Tongji University
[7] Shenzhen Technology University [8] National University of Singapore [9] JD iCity, JD Technology
{wk520529, wuhao2022, duanyifan28}@mail.ustc.edu.cn
bin2003@tongji.edu.cn, Kai.wang@comp.nus.edu.sg
msyuzheng@outlook.com ,pengxiaojiang@sztu.edu.cn
yuxliang@outlook.com*, angyan@ustc.edu.cn*

## Abstract

Spatio-temporal (ST) prediction plays a pivotal role in earth sciences, such as
meteorological prediction, urban computing. Adequate high-quality data, cou-
pled with deep models capable of inference, are both indispensable and prereq-
uisite for achieving meaningful results. However, the sparsity of data and the
high costs associated with deploying sensors lead to significant data imbalances.
Models that are overly tailored and lack causal relationships further compromise
the generalizabilities of inference methods. Towards this end, we `first` es-
tablish a causal concept for ST predictions, named `NuwaDynamics`, which tar-
gets to identify causal regions in data and endow model with causal reasoning
ability in a two-stage process. Concretely, we initially leverage upstream self-
supervision to discern causal important patches, imbuing the model with gen-
eralized information and conducting informed interventions on complementary
trivial patches to extrapolate potential test distributions. This phase is referred
to as the **discovery** step. Advancing beyond the discovery step, we transfer the
data to downstream tasks for targeted ST objectives, aiding the model in recog-
nizing a broader potential distribution and fostering its causal perceptual capa-
bilities (denoted as **Update** step). Our concept aligns seamlessly with the con-
temporary backdoor adjustment mechanism in causality theory. Extensive experi-
ments on six real-world ST benchmarks showcase that models can gain outcomes
upon the integration of the `NuwaDynamics` concept. `NuwaDynamics` also can
significantly benefit a wide range of changeable ST tasks like extreme weather
and long temporal step super-resolution predictions. Our codes are available at
https://github.com/easylearningscores/NuwaDynamics.

## 1 Introduction

Modern deep learning (DL) approaches have demonstrated promising outcomes in various dynami-
cal systems in natural and social science fields like weather forecasting (Schultz et al., 2021; Pathak
et al., 2022; Bi et al., 2022), rapid fire progression (Tam et al., 2022), intelligent transportation (Kaf-
fash et al., 2021; Jin et al., 2023). Such astonishing achievements primarily stem from two crucial
factors. (1) With the development of computer science, a vast amount of data from Earth systems
is continuously being acquired (Chen et al., 2022b; Liu et al., 2023a). These ever-growing, massive
datasets, with diverse sources, provide the impetus for data-hungry deep models, making learning
from data possible. (2) Continual breakthroughs in deep learning algorithms and models enable us
to effectively adapt to diverse specific scenarios, resulting in state-of-the-art performances.

In general, deep learning (DL) provides an efficient optimization framework for automatically and
dynamically extracting intrinsic patterns from continuous observable processes. Unlike classical

---

*Yang Wang and Yuxuan Liang are the corresponding authors. † Kun Wang and Hao Wu contribute equally
to this paper.

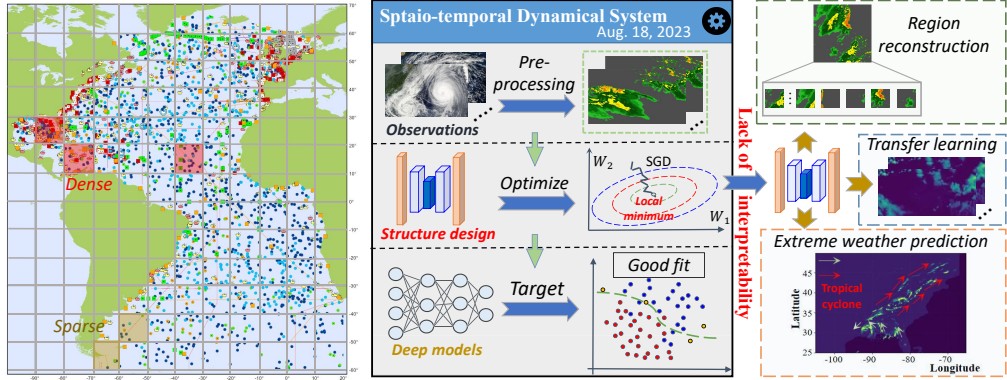

Figure 1: (a) The distribution of uneven ocean system in the Atlantic Ocean. The points represent different types of systems and we mark some red and yellow areas to highlight the imbalance issue of the sensor system. (b) When constructing an ST system, the lack of interpretability often results in limited predictive capabilities for specific scenarios.

dynamic systems, which are primarily derived from first principles (Pryor, 2009; Bürkle et al., 2021) and involve high computational costs, DL approachs often sacrifice an explicit understanding of physical rules. Instead, it resorts to large-scale observable data and captures implicit patterns that serve as substitutes for physical laws. Moreover, these patterns can be understood as spatio-temporal (ST) correlations, and these deep models can be further regarded as ST dynamical systems.

Though promising, in the context of data-driven dynamical systems research, there still exist some clouds on the horizon. 1) High-quality/resolution observational data is relatively scarce, and the cost of training with such data is exceptionally high; the distribution of sensors across various regions on Earth is significantly uneven, with many areas unable to benefit from them effectively due to data scarcity (see the example in Fig 1(a)). More cases are provided in Appendix A. Although some efforts have been made to address this problem, such as transfer learning (Wang et al., 2018a) and active learning (Ren et al., 2021), the data-driven approach still lacks interpretability, resulting in a lack of generalization ability in the transfer process and poor performance in certain extreme scenarios, e.g., cyclone tracking, and turbulence sensing. This remains a common challenge for deep models as shown in Fig 1(b). 2) Customized designs for specific tasks endow the models with specialized capabilities and high performance, however, the complicated designs make the model difficult to generalize. Consequently, unlike enormous public model zoos in NLP and CV realms, each spatio-temporal dynamical system is primarily focused on performing a specific scene task and lacks the ability to transfer knowledge from a higher perspective. For instance, using infrared meteorological data from one region for rainfall prediction in another region.

In this paper, we propose a novel research line for the first time, namely, **causal spatio-temporal dynamics**, aiming to provide an interpretable paradigm for future large-scale ST dynamical systems. Guided by the currently prevailing technique of causal invariant learning (Arjovsky et al., 2019; Sagawa et al., 2019; Rosenfeld et al., 2020; Chang et al., 2020; Liu et al., 2022), our primary objective is to *reveal the inherent correlations in available high-quality measurement data*, thus providing interpretability for complex ST problems in dynamical systems such as representation learning and transfer learning. By leveraging the causal patterns inherent in the limited data, our approach bolsters the reliability of processes such as representation learning and transfer learning. Additionally, our method subtly performs data augmentation on sparse and extreme scenarios, thereby enhancing the model's ability to perceive and understand such circumstances. This enables the extraction of causal features to aid in downstream tasks, leading to an improved, streamlined model performance.

**Uncovering Causal Correlations.** We present the first attempt to introduce the concept of causality to ST dynamical systems by establishing a novel philosophical framework termed `NuwaDynamics`. Briefly put, our objective is to *inject the invariant characteristics and the internal causal patterns within the data from upstream self-supervised tasks, providing a faithful and reliable framework for downstream learning.* Concretely, we decompose our process into two stages – Discovery and Update. The **Discovery** stage aims to answer the question of identifying the latent causal components within observed data, where we introduce self-supervised tasks to the upstream ST data reconstruction. Using the popular Vision Transformer architecture (Khan et al., 2022), we first patchify the

observations at each time step, and then utilize attention maps to localize crucial regions (Selvaraju et al., 2017; Wang et al., 2020a; Jiang et al., 2021a). These localizations are combined with existing pixel-space visualizations to create causal patches.

Going beyond the above process, the **Update** stage endeavors to evolve the downstream tasks into our causal ST model. By appropriately augmenting non-causal patches (*i.e.*, environmental patches), we are in effect generating different randomly deformed copies of the original data. As a result, the model is exposed to a broader distribution of latent data and extreme scenarios, offering insights with a causal perspective for downstream tasks. This process can be further understood as backdoor adjustment in the causal theory (Pearl, 2009; Pearl & Mackenzie, 2018). We believe that such insights will open avenues for future research on learning ST systems and their real-world applications.

Our contributions can be summarized in the following four aspects:

- In this paper, we present a causal and resilient philosophy (`NuwaDynamics`) for modeling spatio-temporal systems with the first shot. Leveraging the causal theory with good interpretability, `NuwaDynamics` allows the model to see a broader potential distribution of data, ensuring the model's outstanding performance across a wide range of downstream tasks.

- In its elegant simplicity, `NuwaDynamics` identifies causal features in its first stage and then refines the model into a causal form. It aspires to master data invariance through upstream self-supervised training, offering a more tailored and reliable foundation for specific downstream tasks.

- `NuwaDynamics` can benefit many existing frameworks on various tasks. For some longstanding challenging issues like extreme weather perception (e.g., hurricanes, high-resolution precipitation), it effectively aids models in achieving perfection in detail.

- We evaluate our framework using eight state-of-the-art models as backbones on six diverse benchmarks, including weather, human motion, fire evolution, pollution diffusion, *etc*. Empirical results show that our concept helps existing models achieve better results in ST representation learning, long-range super-resolution forecasting, and transfer learning. Even in extreme events featured by data scarcity, `NuwaDynamics` has showcased a remarkable ability to capture intricate details.

## 2 PRELIMINARIES

**Spatio-Temporal Forecasting Models** mostly fall into three categories: those grounded in CNNs (Oh et al., 2015; Mathieu et al., 2015; Tulyakov et al., 2018), those rooted in RNNs (Srivastava et al., 2015; Villegas et al., 2017; 2018; Kim et al., 2019; Wang et al., 2022b; Tan et al., 2023), and an assortment of other architectures which include hybrid models (Weissenborn et al., 2019; Kumar et al., 2019) and transformer-centric designs (Dosovitskiy et al., 2020; Gao et al., 2022b; Bai et al., 2022; Wu et al., 2023b). Notably, there are models that leverage graph neural networks (GNNs) primarily for graph data management (Sun et al., 2020; Wang et al., 2020b; Jiang et al., 2021b; Wang et al., 2022a). However, these are outside the scope of our research as we focus on the visualization of ST observational data (Chen et al., 2022b; Veillette et al., 2020). Within our research, we formulate ST observations as an ST sequence $[\mathcal{X}_t]_{t=1}^T, \mathcal{X}_t \in \mathbb{R}^{H \times W \times C_{in}}$. Based on these observations, we aim to parallelly predict the $K$-step-ahead future $[\mathcal{Y}_{T+t}]_{t=1}^K, \mathcal{X}_t \in \mathbb{R}^{H \times W \times C_{out}}$, where $H$ and $W$ denote the number of spatial grids with $C_{in}$ or $C_{out}$-dimensional observations.

**Causal Inference** has garnered considerable attention in the realm of deep learning (Zhang et al., 2020a; Woo et al., 2022; Zheng et al., 2021; Arjovsky et al., 2019) in recent years. Conceptually, causal inference (Pearl et al., 2000; Pearl, 2009) focuses on uncovering the causal relationships between variables, aiming to achieve stable and robust learning and inference. Central to the idea of `discovery` is the commitment to identifying spurious correlations (Geirhos et al., 2018; Sagawa et al., 2019; Koh et al., 2021; Gulrajani & Lopez-Paz, 2020). Discovering spurious correlations exposes model biases that can adversely affect generalization (Wu et al., 2023c). Recently, many techniques (Selvaraju et al., 2017; 2016; Luo et al., 2020; Ying et al., 2019) have been employed for crucial causal features perception. These techniques are sufficiently versatile, exhibiting strong influence across various scenarios (Wu et al., 2023c; Fu et al., 2020; Sui et al., 2022; Zhang et al., 2020b). In this paper, we introduce a novel ST causal framework; by leveraging attention techniques, we can more effectively identify causal regions in observations from the upstream task, providing a more robust foundation for downstream tasks.

**Vision Transformer (ViT) Pruning.** Our work closely resembles the popular ViT image token pruning techniques (Dosovitskiy et al., 2020). However, `NuwaDynamics` emphasizes identifying

important tokens rather than performing token pruning. These endeavors (Rao et al., 2021; Pan et al., 2021; Yuan et al., 2021; Xu et al., 2022) attempt to distinguish how informative a token is by using classification token `[Cls]` as the guideline. However, `NuwaDynamics` upstream focuses on pre-training to reconstruct patches, aiming to discover causal patches, without involving `[Cls]` tokens. Hence, traditional ViT pruning approaches may not be suitable for our framework.

## 3 METHODOLOGY

In this section, we systematically introduce our `NuwaDynamics` framework. We begin with an example of causality, which serves as the motivation behind our approach in Sec 3.1. Subsequently, we provide a detailed account of our algorithmic process, encompassing the upstream self-supervised tasks in Sec 3.2 and the specifics of the downstream spatio-temporal tasks in Sec 3.3.

### 3.1 MOTIVATION EXAMPLES

Let us first consider an example as shown in the upper part of Fig 2. If we only focus on correlations between exercise duration and cholesterol levels, we may observe that longer exercise durations are potentially linked to higher cholesterol levels, which contradicts common sense. Merely using a framework to model this is very likely to produce incorrect conclusions. However, this issue arises because we haven't taken the variable "age" into account. In reality, age affects both exercise duration and cholesterol levels, resulting in the observed data pattern. We hope to uncover more of the data's latent distribution through increased data augmentation, aiming to mitigate such issues. We also provide a more quantitative example in Appendix B.

### 3.2 DISCOVERY SPURIOUS CORRELATIONS

Based on the above motivations, we take a causal look at the ST data-generating process and formalize the principle of identifying causal and non-trivial regions in input observations, which guides our **discovery** strategy (left hand in Fig 3). Naturally, we need a universal mechanism to inspect the causal and spurious regions in the input image. We resort to currently popular ViT tools (Dosovitskiy et al., 2020) which decompose images into patches of equal size and attempt to locate essential patches. However, previous ViT pruning techniques (Pan et al., 2021; Yuan et al., 2021; Xu et al., 2022) have primarily focused on classification tasks and do not transfer well to our spatiotemporal prediction scenarios. Hence, for the first time, we propose an upstream self-supervised reconstruction task and aim to identify potential causal regions during the reconstruction process. Vision transformer first splits the image $\mathcal{X} \in \mathbb{R}^{H \times W \times C_{in}}$ into $L = HW/p^2$ non-overlapping patch tokens and embedding it into a $D$ dimension feature space. Then all tokens are added with a learnable position encoding and then fed into a stacked transformer block:

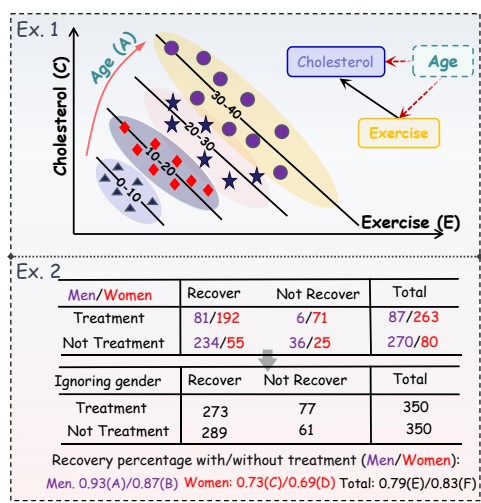

Figure 2: The motivation of our proposal.

$$\mathcal{X}_{\text{MHSA}} = \mathcal{X} + \text{MHSA}\left(\text{LN}\left(\mathcal{X}\right)\right), \quad \mathcal{X}_{\text{FFN}} = \mathcal{X}_{\text{MHSA}} + \text{FFN}\left(\text{LN}\left(\mathcal{X}\right)\right), \tag{1}$$

where MHSA denotes multi-head self-attention (Vaswani et al., 2017); FFN and LN represent a feed-forward network and layer normalization, respectively. In this circumstance, the input is mapped to query, key and value matrices, *i.e.*, $Q, K, V \in \mathbb{R}^{L \times D}$. Then, we can calculate the attention weights $\text{Att} \in \mathbb{R}^{L \times L}$ by using a softmax function, and the attention weight of the $i$-th patch towards the $j$-th patch can be represented as:

$$\alpha_{i,j} = \text{softmax}\left(q_i k_j^T / \sqrt{D_H}\right) \in \text{Att}, \quad \text{where } q_i \in Q, \ k_j \in K. \tag{2}$$

The input are sliced into $\Lambda$ attention heads, and here $D_H = D/\Lambda$ is the feature dimension. Here, we employ attention weight to describe patch importance without introducing any additional variables or hyperparameters. However, for each row of the attention map, we have $\alpha_{i,*} = \sum_{j=1}^{L} \alpha_{i,j} = 1$,

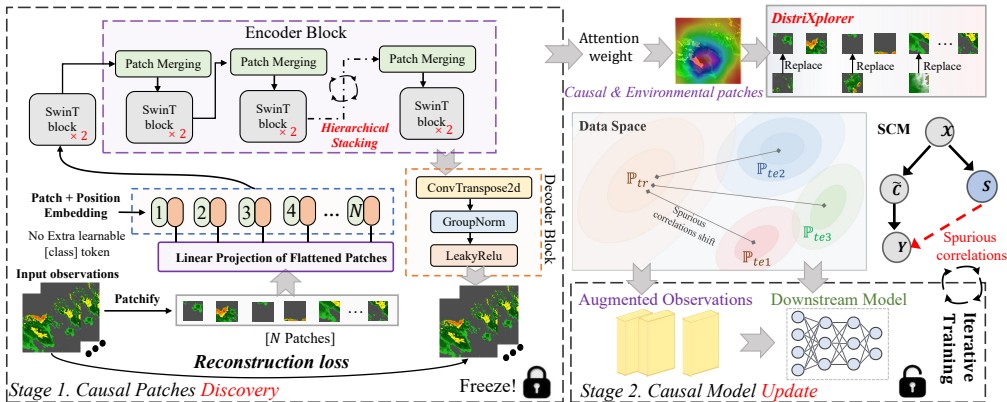

Figure 3: The details of `NuwaDynamics`, in which consists of **Discovery** and **update** two stages. For ease of understanding, we use Swin Transformer as the upstream model.

in which we cannot distinguish the importance of each patch. Hence, we resort to the column score of the attention map and calculate the summation of column attention weights:

$$\alpha_{*,j} = \sum_{i=1}^{L} \alpha_{i,j}, \qquad \alpha_{*,j}^{mean} = \sum_{h=1}^{H} \alpha_{*,j}^{h}/H. \tag{3}$$

Clearly, $\alpha_{*,j}$ exhibits the total attention weights of other tokens to the current token, which can be sufficient to indicate the importance of the current token. $\alpha_{*,j}^{mean}$ represents the average importance and the weight of $j$-th patch across $\Lambda$ heads in multi-head attention. Then we move forward to calculate the normalized importance $I_j$ of each patch and select the parts set of smallest values (Note as $\mathcal{M}$) as environmental patches as:

$$I_j = \alpha_{*,j}^{mean}/\sum_{j=1}^{L} \alpha_{*,j}^{mean} \in I, \qquad \mathcal{M} = \text{index}_{\tilde{c}}\{I_{\tilde{c}}|I_{\tilde{c}} = arg\min_{j \in [1,L]} I_j\}. \tag{4}$$

We employ the saliency map $\mathcal{M} = \{0, 1\} \in \mathbb{R}^L$ by sampling the smallest elements in $I$. In this way, we can identify the causal patches $z_{\tilde{c}}$ and the complementary environments $z_s$. In general, due to the unstable nature of spurious attributes (Wu et al., 2023c), the test distribution $\mathbb{P}_{te}$ is often different from the training settings, *i.e.*, $\mathbb{P}_{te} \neq \mathbb{P}_{tr}$. `NuwaDynamics` is dedicated to enhancing model perception of the underlying essence of data. This not only facilitates representation learning and transfer learning tasks but also addresses the challenges posed by data scarcity.

**DistriXplorer.** Recall that the causal theory (Pearl, 2009; Bunge, 2017) attributes the model's weak generalization capability to the distribution shift of spurious associations, namely, the environmental part. We resort to causal intervention to forcibly assign values to environmental patches. Towards this end, we design a **DistriXplorer** to modify the environmental patches, aiming to enhance scenarios with observable environmental patches. However, intervening at the patch level is complex. Existing interventions primarily focus on the class level (Wu et al., 2023c) and the graph realm (Feng et al., 2021). Interestingly, the characteristics of a particular patch are often influenced by its spatial neighboring and temporal historical patches. Guided by this property, we elaborate an ST mixup DistriXplorer. Concretely, we sample spatial neighboring and temporal historical patches to mix up and generate different random deformed copies:

$$z_S^t = \sum_{i=1}^{\mathcal{O}} \lambda_{ner\_i} \sum_{j=1}^{t} \beta^{t+1-j} \cdot z_{ner\_i}^j, \quad \text{where } \lambda_{ner\_i} = I_{ner\_i}^t/\sum_{i=1}^{\mathcal{O}} I_{ner\_i}^t \tag{5}$$

Here $z_S^t$ denotes the environmental patches at $t$ points. $\mathcal{O}$ represents the number of causal patches among the neighbors. $\lambda_{ner\_i}$ and $\beta$ are the weight allocated for spatial neighboring patches and temporal decaying coefficient. Going beyond interventions, we randomly sample the surrounding patches with a probability of $\Omega \sim \text{Uniform}(0, 1)$ to generate multiple ordered sequences for next training. In this way, the downstream model can learn patterns for adjusting the environment to improve the generalizability (We summarize generative methods that can be integrated into our augmentation in Future Work, here we choose Mixup for trade-off of efficiency and performance).

**Causal Support of `NuwaDynamics`.** Drawing from the causal theory, we construct a Structural Causal Model (SCM) (Pearl, 2009) by examining four variables: input observations $\mathcal{X}$, ground-truth $\mathcal{Y}$, causal patches in $\mathcal{X}$ denoted as $\tilde{C}$, and the confounder (i.e., environment) represented by $S$. Then we can depict the causal relationships among them by:

- $\tilde{C} \leftarrow \mathcal{X} \rightarrow S$. The input $\mathcal{X}$ consists of two disjoint parts $\tilde{C}$ and $S$.
- $\tilde{C} \rightarrow \mathcal{Y} \dashleftarrow S$. $\tilde{C}$ is the only endogenous parent to determine the ground-truth $\mathcal{Y}$. However, in practical scenarios, $S$ is simultaneously used for predicting $\mathcal{Y}$, leading to spurious associations.

In general, a model $\mathcal{F}_\emptyset$ trained with Empirical Risk Minimization (ERM) often falls short of generalizing to the test data $\mathcal{D}_{te} \sim \mathbb{P}_{te}$. These distribution shifts are triggered by changes in the environmental patches. Therefore, it is imperative to address the confounding effect exerted by the environmental confounder. As shown in the right panel of Fig 3, we employ causal intervention to assist the downstream models in perceiving a broader range of test distributions, *i.e.*, $\mathbb{P}_{te1}, \mathbb{P}_{te2}$, *etc.* Our framework exploits **do-calculus** (Pearl et al., 2000) on variable $\tilde{C}$ to remove the backdoor patch $S \dashrightarrow \mathcal{Y}$ by estimating $P(\mathcal{Y}|do(\tilde{C})) = P_m(\mathcal{Y}|\tilde{C})$:

$$P_m(\mathcal{Y}|\tilde{C}) = P\left(\mathcal{Y}|do\left(\tilde{C}\right)\right) = \sum\nolimits_{i=1}^{\mathcal{T}} P\left(\mathcal{Y}|\mathcal{X}, S_i\right) P\left(S = S_i\right) \tag{6}$$

where $\mathcal{T}$ denotes the number of environments. $S_i$ denotes the $i$-th environmental variable. The environmental enhancement at upstream of Nuwa aligns well with the backdoor adjustment theory, thereby effectively exploring the potential test environment distributions. Detailed proofs are provided in Appendix H.

### 3.3 UPGRADING TO CAUSAL INFERENCE: A NEW ERA IN MODELING

Typically, downstream models can be categorized into transformer and non-transformer classes. For the transformer class, ensuring consistency between the upstream and downstream models allows for rapid parameter transfer, facilitating quicker optimization of the downstream model. On the other hand, for non-transformer architectures, we employ transfer-augmented data to optimize and update the downstream models, advancing their causal perception capabilities. We store the intervention data described in Sec 3.2 in the spatio-temporal bank $\text{ST}(t)$. In downstream tasks, we retrieve the data from the bank, ensuring the consistency of prediction labels between the intervention data and the original data for parallel training. However, as each timestamp has its corresponding environmental patches, denoted by $\xi^t$ for the number of environmental patches at time $t$, theoretically, $2^{\sum_t \xi^t}$ prediction sequences can be constructed. This poses a significant, or even intractable, computational burden on the model. In the ST scenario, we argue that historical data closer to the current moment potentially have a greater influence. Therefore, we propose a temporal Gaussian decay sampling method to identify more influential data, aiming to enhance the model's generalization ability while reducing its computational burden:

$$\mathcal{G}\left(T, \sigma^2\right) = \frac{1}{\sigma\sqrt{2\pi}}\exp\left(-\frac{(x-T)^2}{2\sigma^2}\right) \quad \text{X} = \left[\mathcal{X}_{(\text{ST}(t),\, \mathcal{G}(t,\sigma^2))}\right]_{t=1}^{T}. \tag{7}$$

We use the current moment $T$ as the mean value, with variance $\sigma^2$ to control the sampling ratio. In this way, we construct our training data as $X$. Details can be found in Appendix I.

## 4 EXPERIMENTS

In this section, we present empirical results to demonstrate the effectiveness of `NuwaDynamics` framework. The experiments aim to investigate the following research questions:

- Does `NuwaDynamics` enhance performance prediction for existing ST backbones?
- Does `NuwaDynamics` outperform on specific challenging tasks?
- Can the acquisition of feature invariance enhance model generalization?
- In the context of rare and extreme events, how effective is `NuwaDynamics` at detection?

### 4.1 EXPERIMENTAL SETTINGS

We conduct extensive experiments to evaluate the effectiveness of `NuwaDynamics`. We implement different backbones using Pytorch and leveraging the A100-PCIE40GB as support. We train all models with Adam optimizer and learning rate as 0.01. More detail can be found in Appendix C.

**Datasets & Backbones** We extensively evaluate our proposal on five benchmarks across diverse research domains, including TaxiBJ+ (Liang et al., 2021), KTH (Schuldt et al., 2004), SEVIR (Veillette et al., 2020), RainNet (Ayzel et al., 2020), PD, and FireSys (Chen et al., 2022a). Specifically, TaxiBJ+ tackles urban traffic, KTH focuses on human kinetics, SEVIR analyzes extreme weather, RainNet forecasts precipitation, PD simulates pollutant dispersion, and FireSys monitors wildfires.

Table 1: Performance comparison on different backbones, where "Ori" refers to the backbones, and "+NuWa" indicates the performance after incorporating `NuwaDynamics`. All experimental results are the average of **five runs** and the red font indicates the optimal value. Except for PD, which is $6 \rightarrow 6$, all others are $10 \rightarrow 10$.

| Backbone ($10 \rightarrow 10$) | Metric | TaxiBJ+ | | KTH | | SEVIR (CSI-M*) | | RainNet | | PD ($6 \rightarrow 6$) | | FireSys | |
|---|---|---|---|---|---|---|---|---|---|---|---|---|---|
| | | Ori | +NuWa | Ori | +NuWa | Ori | +NuWa | Ori | +NuWa | Ori | +NuWa | Ori | +NuWa |
| *The upstream architecture is Transformer based and maintain consistency between upstream and downstream structures.* | | | | | | | | | | | | | |
| ViT [2020] | MAE | 3.48 | 2.27 | 59.32 | 34.56 | 37.07 | 46.88 | 0.78 | 0.74 | 83.45 | 24.70 | 3.21 | 3.09 |
| | MSE | 0.16 | 0.07 | 57.88 | 35.43 | 4.53 | 3.16 | 0.23 | 0.19 | 8.99 | 2.45 | 8.27 | 8.19 |
| | $\Delta$ | | 0.09 | | 22.45 | | 1.37 | | 0.04 | | 6.51 | | 0.08 |
| SwinT [2021] | MAE | 3.22 | 2.18 | 55.44 | 33.45 | 38.22 | 45.68 | 0.67 | 0.66 | 79.53 | 26.38 | 2.98 | 2.76 |
| | MSE | 0.21 | 0.11 | 52.38 | 33.11 | 4.37 | 2.84 | 0.22 | 0.19 | 8.47 | 3.15 | 7.96 | 7.65 |
| | $\Delta$ | | 0.10 | | 19.27 | | 1.89 | | 0.03 | | 5.32 | | 0.31 |
| Rainformer [2022] | MAE | — — | — — | 80.32 | 40.77 | 36.68 | 46.88 | 1.21 | 1.17 | 81.23 | 30.54 | 4.65 | 4.55 |
| | MSE | — — | — — | 77.99 | 40.75 | 4.02 | 3.38 | 0.30 | 0.21 | 8.63 | 2.51 | 11.27 | 10.72 |
| | $\Delta$ | — — | — — | | 37.24 | | 0.64 | | 0.09 | | 6.12 | | 0.55 |
| Earthformer [2022b] | MAE | — — | — — | 52.37 | 42.91 | 44.21 | 46.33 | 1.98 | 1.54 | 73.24 | 30.78 | 1.97 | 1.57 |
| | MSE | — — | — — | 48.65 | 37.21 | 3.88 | 2.96 | 0.20 | 0.19 | 7.32 | 2.44 | 5.17 | 4.94 |
| | $\Delta$ | — — | — — | | 11.44 | | 0.92 | | 0.01 | | 4.88 | | 0.23 |
| *The upstream architecture is ViT and downstream does not specify a particular model architecture.* | | | | | | | | | | | | | |
| ConvLSTM [2015] | MAE | 5.52 | 3.27 | 128.33 | 53.10 | 41.93 | 44.88 | 3.98 | 3.64 | 100.44 | 58.39 | 11.21 | 10.97 |
| | MSE | 0.33 | 0.27 | 126.32 | 89.35 | 3.84 | 3.17 | 0.49 | 0.30 | 10.98 | 5.47 | 17.22 | 16.43 |
| | $\Delta$ | | 0.06 | | 36.97 | | 0.67 | | 0.19 | | 5.51 | | 0.79 |
| PredRNN-V2 [2022b] | MAE | 4.33 | 3.25 | 51.38 | 40.37 | 40.83 | 44.99 | 2.67 | 2.43 | 95.43 | 72.77 | 4.32 | 3.97 |
| | MSE | 0.27 | 0.20 | 51.36 | 45.76 | 3.98 | 3.17 | 0.41 | 0.33 | 9.65 | 7.35 | 5.87 | 4.53 |
| | $\Delta$ | | 0.07 | | 5.60 | | 0.81 | | 0.08 | | 2.30 | | 1.34 |
| E3D-LSTM [2018b] | MAE | 4.25 | 3.27 | 86.37 | 52.98 | 40.56 | 45.38 | 3.88 | 3.72 | 100.23 | 78.34 | 4.98 | 4.65 |
| | MSE | 0.29 | 0.25 | 87.69 | 59.49 | 4.37 | 3.89 | 0.38 | 0.29 | 10.34 | 7.35 | 8.76 | 8.12 |
| | $\Delta$ | | 0.04 | | 28.20 | | 0.48 | | 0.09 | | 2.99 | | 0.64 |
| SimVP [2022a] | MAE | 3.07 | 2.56 | 43.39 | 33.98 | 45.98 | 47.09 | 1.27 | 1.02 | 50.93 | 31.55 | 1.98 | 1.54 |
| | MSE | 0.14 | 0.07 | 40.93 | 32.89 | 3.44 | 2.92 | 0.28 | 0.20 | 5.48 | 3.24 | 2.65 | 2.42 |
| | $\Delta$ | | 0.07 | | 8.04 | | 0.52 | | 0.08 | | 2.24 | | 0.23 |

In our predictions, we simultaneously utilize the past 10 images to forecast the next 10. Additionally, due to the larger resolution of the PD dataset, we adopt a $6 \rightarrow 6$ approach. Given that the upstream framework employs a Transformer-based architecture, we ensure the downstream structure both emulates and differentiates from the upstream one in order to validate the universality of our algorithm. Concretely, we use Transformer-based models as our backbone, such as ViT (Dosovitskiy et al., 2020), SwinT (Liu et al., 2021), Rainformer (Bai et al., 2022) and Earthformer (Gao et al., 2022b), as well as non-Transformers such as ConvLSTM (Shi et al., 2015), PredRNN-V2 (Wang et al., 2022b), E3D-LSTM (Wang et al., 2018b) and SimVP (Gao et al., 2022a). All Transformers had 12 encoder blocks, while non-Transformers used Transpose Conv2d for upsampling. This evaluation aims to clarify the efficacy of each architecture in managing `NuwaDynamics'` complexities, laying a solid foundation for future model refinement. More detail can be found in Appendix C.

**Measurement metric.** We delve into the metrics used by evaluation methods. Concretely, we train backbones with mean squared error (MSE), and use mean absolute error (MAE), MSE and structural similarity index measure (SSIM) as evaluation metrics. Specifically, for the SEVIR dataset, we incorporate the CSI index (Ayzel et al., 2020) to replace MAE as a primary metric for comparison. For MAE and MSE, we use ↓ indicates better performance, and higher value (↑) denotes better results for SSIM and CSI-M. More details are placed in Appendix C.

## 4.2 ASSESSING THE EFFICACY OF NUWADYNAMICS (RQ1)

As a preparation, we selected both Transformer and non-Transformer architectures. For the Transformer architecture in our upstream tasks, we ensure that the sequence lengths of the upstream reconstruction tasks and downstream prediction tasks are consistent, allowing for direct model parameter transfer. For non-transformer architectures, we only transfer the data to train the downstream models. Marker ■ and ■ denote the decrement in MSE and the variation in CSI-M, respectively. We summarize the results in Tab 1, from the experimental results, We make the following **Obs**ervations:

**Obs 1. +Nuwa consistently outperforms without `NuwaDynamics` concept.** As shown in Tab 1, we can easily observe that upon integrating the `NuwaDynamics` concept into the model (+Nuwa), there were consistent improvements in performance. This is evident from the reductions observed in both MSE and MAE. Specifically, for complex spatiotemporal data such as PD, introducing +Nuwa can yield significant benefits: a range of $4.88 \sim 6.51$ descents in Transformer scenarios and $0.08 \sim 7.35$ in non-transformer scenarios across 8 different backbones based on MSE metrics.

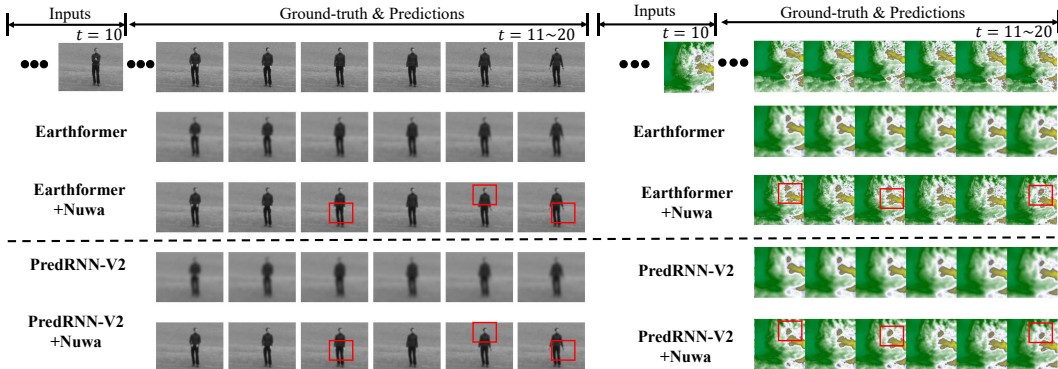

Figure 5: Visualization on KTH & SEVIR. For simplicity, we display the results of the last 6 frames.

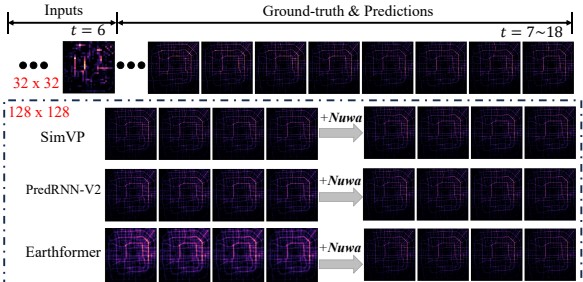

Figure 4: Visualizations on backbones and +Nuwa. For layout convenience, we only display the last six frames and showcase complete results in Appendix E.

| Settings (TaxiBJ+) | | Output Sequence | | |
|---|---|---|---|---|
| Backbone | Input Seq (6) | $\to 6$ | $\to 8$ | $\to 12$ |
| SimVP | w/o Nuwa | 98.67 | 96.43 | 94.32 |
| | + Nuwa | 99.12 | 97.77 | 95.12 |
| PredRNN-V2 | w/o Nuwa | 94.53 | 93.41 | 89.77 |
| | + Nuwa | 96.89 | 94.54 | 91.21 |
| Earthformer | w/o Nuwa | 87.12 | 85.44 | 76.56 |
| | + Nuwa | 89.92 | 86.43 | 77.12 |

Table 2: Model performances on three backbones under w/o and + Nuwa conditions. We set the input sequence as 6 frames and the predictive length as 6, 8, 12 with SSIM performances.

**Obs 2. NuwaDynamics demonstrates remarkable adaptability across a myriad of spatiotemporal scenarios.** NuwaDynamics has been validated across a wide range of ST realm, including traffic, human motion, climate, precipitation, *etc.* These real-world datasets repeatedly attest to the robust generalizability of NuwaDynamics. For instance, on climate and precipitation datasets like SEVIR and RainNet, there was an average MSE reduction of approximately 0.13 and 0.91, respectively. Notably, when integrating causal perturbations on the SEVIR dataset under the CSI-M metric, improvements of approximately 7.40 and 3.28 were observed for Transformer and non-transformer architectures, respectively. Further analysis of visualization results are presented in Fig 12.

**Obs 3. NuwaDynamics excels in capturing predictive details.** When attached NuwaDynamics, Earthformer and PredRNN-V2 present the predictions in good sharpness. In KTH, NuwaDynamics could help the model to predict the sharpest sequence compared with original backbones and largely enrich the details for each part of the body, especially for the arms and legs. In SEVIR, the model achieves more reliable predictions on details, with texture information becoming more pronounced.

### 4.3 EVALUATING THE PERFORMANCES ON CHALLENGING TASK (RQ2)

Although the aforementioned experiments have demonstrated the efficacy of NuwaDynamics, validations have been limited. For instance, we have only verified the model under conditions where the temporal lengths and image sizes between upstream and downstream tasks are consistent. To further elucidate the robust adaptability, we choose a more challenging ST task, *i.e.,* the long temporal step super-resolution prediction. Specifically, we selected TaxiBJ+ as the validation benchmark, since it has intricate temporal dynamics. The input images were downsampled to $32 \times 32$, and we utilize the past 6 frames to predict the next 12 frames at a resolution of $128 \times 128$. To accommodate various spatial resolutions and diverse temporal lengths, we employ a spatial sampling module for the downstream framework (Here we choose SimVP, PredRNN-V2 and Earthformer). For a fairer comparison, we only perform data transfer without transferring model parameters. We have placed the setting details in Appendix F. From the Fig 4 and Tab 2, we make observations:

**Obs 4. NuwaDynamics shows great prominence in challenging task.** We find that for long-range super-resolution prediction tasks, all models benefit from Nuwa. As depicted in Fig 4, Earthformer exhibits the most pronounced advantage. In long-distance forecasting scenarios, especially for 12-

time-step predictions, SimVP, PredRNN-V2, and Earthformer all achieve an improvement ranging from 0.56 to 1.44 on SSIM. This further attests to the efficacy of our model.

## 4.4 TRANSFERABILITY OF NUWADYNAMICS (RQ3)

ST transfer learning has long been considered a challenging problem, given its intricate ST correlations (Yao et al., 2020; Wang et al., 2018a). Few works have managed to effectively transfer certain ST patterns to assist another scene. *RegionTrans* (Wang et al., 2018a) takes the first step to achieve inter-city knowledge transfer, and (Yao et al., 2020) designs a differentiable frame for unsupervised transfer learning across multiple ST tasks. In this study, we chose a more complex sense of meteorological prediction as a backdrop to explore whether ST transfer tasks can benefit from `NuwaDynamics`. Concretely, we select two meteorological datasets, RainNet (50GB) and SEVIR (100.6GB), as source data and target data (Note as RainNet ⇌ SEVIR), respectively. Further, we choose SOTA frameworks based on CNN (SimVP), RNN (PredRNN-V2), and Transformer (Earthformer) to systematically validate the feasibility of our framework.

**Obs 5.** `NuwaDynamics` **greatly improves transferability of general models.** From a holistic insight, the model's transfer capability has improved by approximately $1.34 \sim 7.75$ on SSIM. This demonstrates Nuwa's assistance in transfer learning. An intriguing observation was made by us: Earthformer exhibits greater capability than SimVP and PredRNN-V2, further supporting the potential superiority of transformers in transfer learning tasks and effectiveness of our upstream task. We showcase the visualizations of the transfer learning results in Appendix G.

Table 3: Performances under w/o and + Nuwa. Marker ▇ and ▇ denote the RainNet (R) → SEVIR (S), and RainNet (R) ← SEVIR (S) performances. $\Delta$ denotes improvements in SSIM metrics.

|  | **SimVP**, $(10 \rightarrow 10)$ | | | **PredRNN-V2**, $(10 \rightarrow 10)$ | | | **Earthformer**, $(10 \rightarrow 10)$ | | |
|---|---|---|---|---|---|---|---|---|---|
|  | w/o Nuwa | +Nuwa | $\Delta$ | w/o Nuwa | +Nuwa | $\Delta$ | w/o Nuwa | +Nuwa | $\Delta$ |
| R → S | 72.12 | 76.98 | 4.86 | 65.43 | 66.97 | 1.54 | 82.18 | 85.67 | 3.49 |
| S → R | 65.49 | 66.97 | 1.48 | 64.37 | 72.12 | 7.75 | 75.32 | 76.66 | 1.34 |

## 4.5 EXTREME WEATHER FORECASTING OF NUWA (RQ4)

Extreme weather forecasting has always been considered a highly challenging task with significant real-world implications (Bi et al., 2022). Due to the rarity of extreme events, current research often struggles to achieve high fidelity in detailing (Scher & Messori, 2019; Schultz et al., 2021; Keisler, 2022). SEVIR and RainNet contain a vast collection of high-quality extreme events like Storm, Hurricane Florence and Squall. We showcase the visualizations in Fig 6 to illustrate that the enhancements in Nuwa's upstream can aid in better discerning the distribution of potential extreme events, thereby improving perceptual capability. As expected, Nuwa helps the model in capturing the intricate details of extreme weather, achieving exceptional local fidelity.

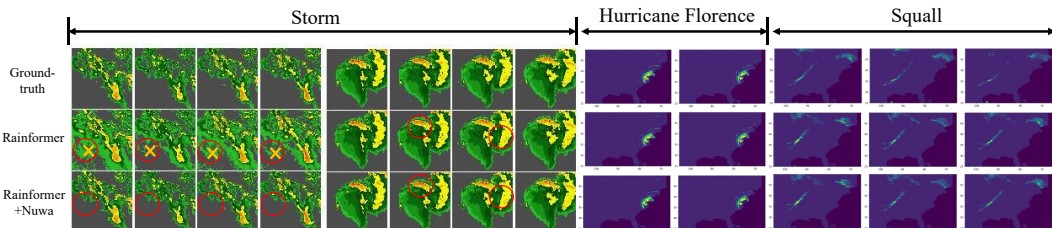

Figure 6: Visualization of storm, hurricane florence, and squall.

## 5 CONCLUSION

In this paper, we present the first attempt to introduce the causality philosophy in ST forecasting tasks. We propose a two-stage causal framework, `NuwaDynamics`, to discover non-trivial regions in data and update the model into causal frameworks. It performs self-supervised learning in upstream reconstruction tasks for intervening in environmental regions and augmentation on trivial part (aligned with the backdoor adjustment). Extensive experiments across six real-world spatiotemporal (ST) benchmarks demonstrate that models enhanced with the NuwaDynamics concept yield improved results. In the future, we plan to explore causal learning on spatio-temporal graphs.

## 6 ACKNOWLEDGEMENT

This paper is partially supported by the National Natural Science Foundation of China (No.62072427, No.12227901), the Project of Stable Support for Youth Team in Basic Research Field, CAS (No.YSBR-005), Academic Leaders Cultivation Program, USTC. The authors also thank Guangzhou-HKUST(GZ) Joint Funding Program (No. 2024A03J0620), National Natural Science Foundation of China (62176165). This work is supported by the National Research Foundation, Singapore under its AI Singapore Programme (AISG Award No: AISG2-PhD-2021-08- 008).

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

# A EXAMPLES

We present the global distribution of PH in 2022, sourced from the average of monthly observations distributed at the Global Disaster Alert and Coordination System (GDACS) over the calendar year (https://www.ocean-ops.org/). From the data (Fig 7), we easily observe a significant imbalance in the PH distribution within global ocean currents. Within each grid area, the number of grids ranging from 0.01 to 0.50 is more than five times greater than the number of grids ranging from 1.01 to 1.50. During data analysis, it becomes challenging to predict extreme regions (e.g., regions with PH > 3.0) using models.

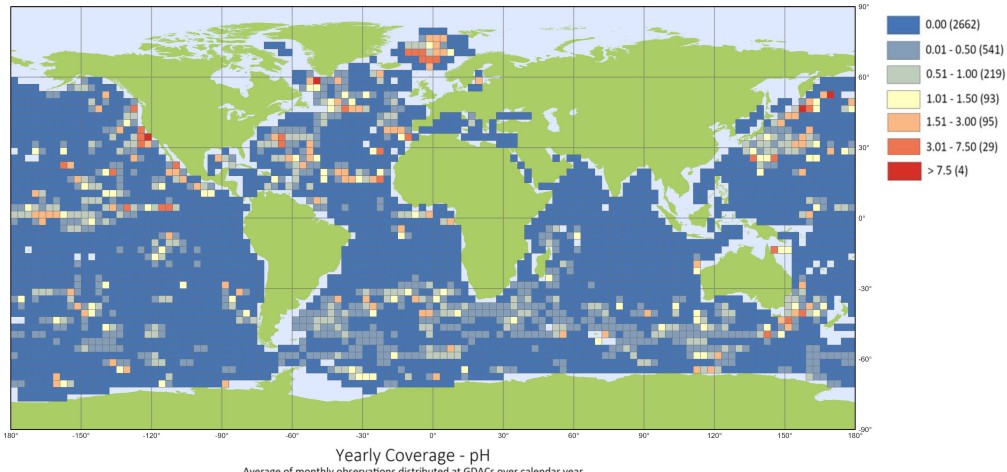

Figure 7: Global distribution of PH in 2022, sourced from the average of monthly observations distributed at the Global Disaster Alert and Coordination System (GDACS) over the calendar year.

Another case of uneven sensor distribution (Fig 8) reveals that the deployment of observation equipment in the Indian Ocean is significantly sparser compared to the Pacific and Atlantic Oceans. This imbalance in observation deployment can lead to challenges in data collection. Leveraging data from regions with a higher density of sensor deployment to guide regions with relatively fewer deployments can provide significant assistance in addressing these challenges.

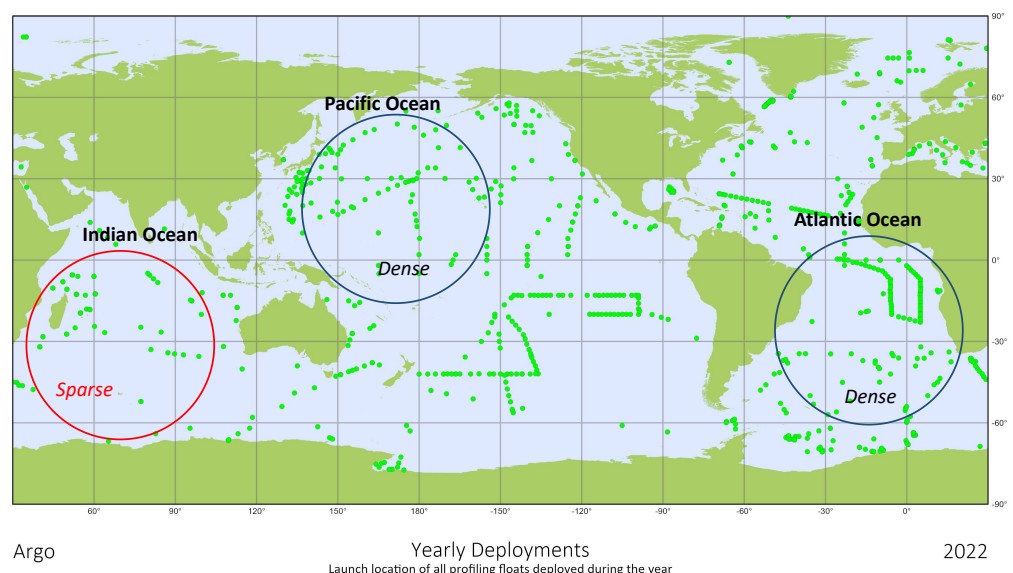

Figure 8: Launch locations for all profile buoy deployments in 2022.

## B    AN QUANTITATIVE EXAMPLE SUPPORT MOTIVATION

We next contemplate a more quantitative example in the lower part of Fig 2. Suppose we are investigating whether a drug aids in recovery from an illness. For males, out of 270/87 who did not take or took the medicine, and the recovery rate among those who took the medicine was 0.93, which is higher than those who did not. A similar phenomenon can be observed in the female cohort (0.73>0.69). However, when we disregard gender, we reach the opposite conclusion, with the recovery rates being 0.79 for those who took the medicine and 0.83 for those who did not. This phenomenon is known as Simpson's Paradox (Pearl et al., 2000), caused by the unobserved gender variable in the aggregate data. By thoroughly traversing the confounder variables, we can effectively mitigate the above issue. This approach forms the crux of our framework and is also referred to as "backdoor adjustment" (Pearl & Mackenzie, 2018; Pearl et al., 2000).

## C    EXPERIMENTAL SETTINGS

**Evaluation metrics.** The Critical Success Index (CSI) (Ayzel et al., 2020; Gao et al., 2022b) serves as a prevalent metric in precipitation forecasting to gauge prediction accuracy. CSI is given by: CSI = $\mathrm{Hits}/(\mathrm{Hits} + \mathrm{Misses} + \mathrm{F.Alarms})$. Here, Hits, Misses, and F.Alarms are the quantities of true positives, false negatives, and false positives. To compute these quantities, we rescale the predicted and true values to lie between 0 and 255 and determine binary classifications using the thresholds [16, 74, 133, 160, 181, 219]

By computing CSI values across various thresholds, we assess the predictive performance of the model, employing the mean CSI-M as a comprehensive evaluation metric. A superior CSI value signifies precise precipitation prediction by the model, whereas an inferior CSI suggests room for enhancement in predictive capability. Thus, the CSI stands as a pivotal metric in precipitation forecasting, offering insight into the model's efficacy and directing refinements in the prediction algorithm.

Hits, Misses, and F.Alarms are important indicators for evaluating the performance of the prediction. Specifically:

- True positive (Hits): Denotes the model's accurate prediction of precipitation occurrence.

- False negative (Misses): Indicates the model's oversight in predicting an actual precipitation event.

- False positive (F.Alarms): Reflects the model's erroneous prediction of precipitation, specifically when precipitation does not materialize.

In predictive modeling, a greater count of Hits, Misses, and F.Alarms implies diminished performance. In precipitation forecasting, our objective is to maximize the number of Hits, concurrently minimizing Misses and F.Alarms, thereby enhancing the accuracy and trustworthiness of the predictions.

**Details of benchmarks.** Here we provide a systematic introduction to the benchmark we used. For a clearer and better understanding, we have placed the statistical characteristics in Tab 4.

Table 4: Dataset statistics. $N\_tr$ and $N\_te$ denote the number of instances in the training and test sets. The lengths of the input and prediction sequences are $I_l$ and $O_l$, respectively.

| Dataset | $N\_tr$ | $N\_te$ | $(C, H, W)$ | $I_l$ | $O_l$ | Interval |
|---|---|---|---|---|---|---|
| TaxiBJ+ | 3555 | 445 | (2, 128, 128) | 10 | 10 | 30 mins |
| KTH | 108717 | 4086 | (1, 128, 128) | 10 | 10 | – |
| SEVIR | 4158 | 500 | (1, 384, 384) | 10 | 10 | 5 mins |
| RainNet | 6000 | 1500 | (1, 208, 333) | 10 | 10 | 1 hour |
| PD | 2000 | 500 | (3, 1400, 1400) | 6 | 6 | 5 seconds |
| FireSys | 2000 | 500 | (3, 128, 128) | 10 | 10 | – |

- **TaxiBJ+**: This dataset encompasses trajectory information sourced from Beijing taxis' GPS, delineated into two distinct channels: inflow and outflow. Furthermore, we've augmented the original dataset by gathering recent trajectory details from Beijing and enhancing the resolution from 32×32 to 128×128, designating it as **TaxiBJ+**.

- **KTH**: This dataset comprises 25 individuals executing six distinct actions: walking, jogging, running, boxing, waving, and clapping. The intricacy of human movements stems from the unique variations individuals exhibit when performing these actions. By analyzing preceding frames, the model can grasp the nuances of human dynamics and anticipate future prolonged postural shifts.

- **SEVIR**: The SEVIR dataset features radar-based readings of vertical accumulation liquid (VIL), captured at 5-minute intervals with a 1 km resolution. This dataset serves as the foundational source for rain and hail detection.

- **RainNet**: This benchmark boasts over 62,400 pairs of top-notch low/high-resolution precipitation maps spanning more than 17 years, primed to facilitate the advancement of deep learning models in precipitation downscaling.

- **Pollutant-Diffusion (PD)**: This data is derived from the computational fluid dynamics (CFD) simulation outcomes related to pollutant dispersion within a designated area. We selected a wind speed of $15m/s$, with the wind direction set to due north, and utilized the centering point as the dynamic data for the pollutant release point.

- **FireSys**: The FireSys dataset encompasses data related to fire observations, where both temporal and spatial trends of fire evolution accurately reflect the progression status in nature.

## D  VISUALIZATIONS TO ANSWER RQ1.

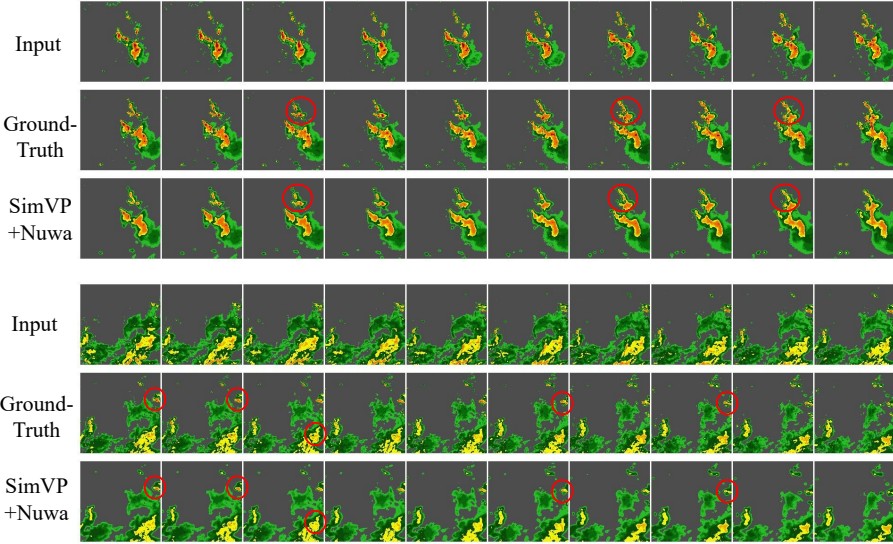

Figure 9: Visualization results of SEVIR under SimVP+Nuwa, we can observe that incorporating NuwaDynamics significantly enhances the model's ability to capture fine-grained details.

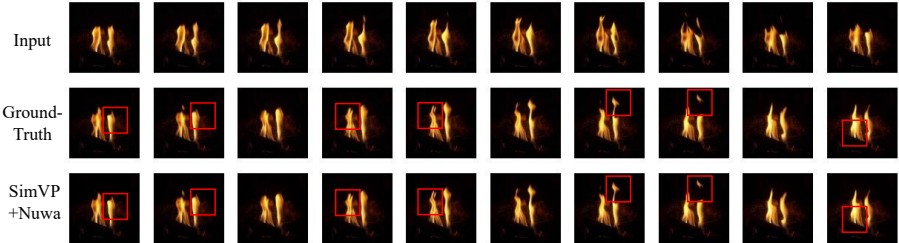

Figure 10: Visualization results of FireSys under SimVP+Nuwa, We find that upon integrating NuwaDynamics, the model's predictive outcomes adeptly capture the edge information of flames, offering a commendable prediction in terms of fine details.

## E  VISUALIZATIONS TO ANSWER RQ2.

In this section, we present the complete visualization results on TaxiBJ+. It's evident that SimVP achieves the best visual details in prediction, while Earthformer's visualization performance is inconsistent. However, when enhanced with Nuwa, all models achieve improved visualization outcomes. This improvement is most notably observed in Earthformer's results.

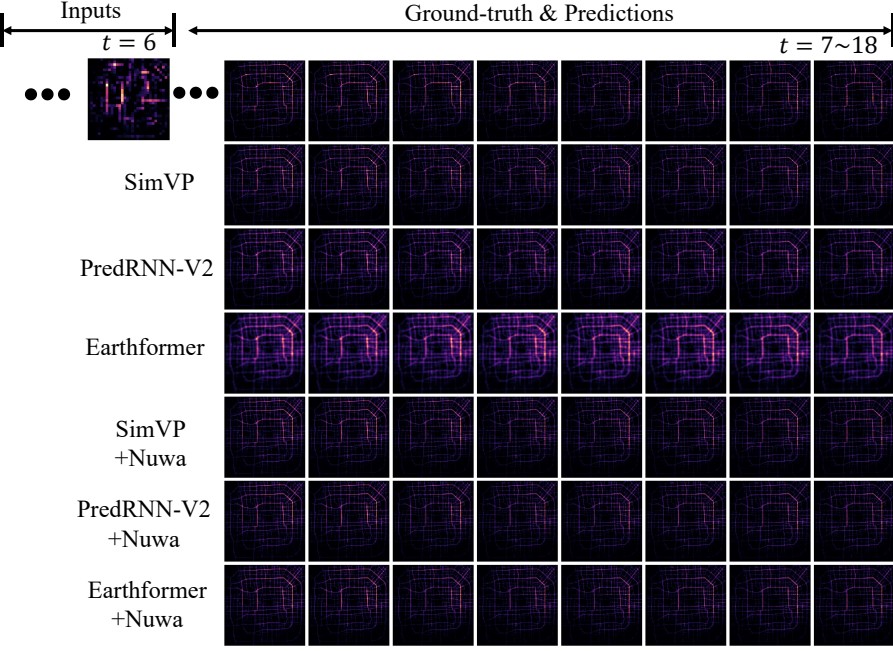

Figure 11: Visualization results of TaxiBJ+ under SimVP+Nuwa, PredRNN-V2+Nuwa and Earthformer+Nuwa. Here we showcase the last eight frames for ease of understanding.

## F  EXPERIMENTAL SETTINGS ON LONG TEMPORAL STEP SUPER-RESOLUTION PREDICTION

In this section, we have delved into scenarios where the input-output dimensions of pre-trained models differ from those of downstream tasks. We addressed this in two parts: (1) handling low-resolution data inputs, and (2) managing varying prediction lengths. Specifically, the details are as follows:

**Spatial Upsampling in Spatio-temporal Data**  The spatio-temporal upsampler contains a specially designed upsampling module aimed at enhancing the spatial resolution of time series data. Given a five-dimensional input tensor $x$ with the shape $B \times T \times C \times H \times W$, where $B$ represents the batch size, $T$ denotes the number of time steps, $C$ stands for the channel count, and $H$ and $W$ respectively describe the height and width of the feature map. This tensor is initially reshaped into a four-dimensional form as shown by $x_{reshape} = reshape(x, (B \times T, C, H, W))$. Subsequently, through a transpose convolution operation with a stride of 4, a kernel size of 4, and zero padding, the spatial dimensions of the feature map are expanded, yielding $x_{upsampled} = \uparrow_{4,4,0} (x_{reshape})$, with the resulting dimensions being $4H \times 4W$. Ultimately, the output feature map $x_{final}$ is reshaped back into its original five-dimensional shape, expressed as $x_{final} = reshape(x_{upsampled}, (B, T, C, 4H, 4W))$. In summary, the entire upsampling process can be represented as $x_{final} = reshape(\uparrow_{4,4,0} (reshape(x, (B \times T, C, H, W))), (B, T, C, 4H, 4W))$, thereby facilitating a precise transformation from low to high resolution.

**Adaptive Temporal Forecasting with Autoregressive Process**  In the context of time series forecasting using a CNN-based method with an input tensor of dimensions $B \times T \times C \times H \times W$, there

exists a challenge in flexibly extending the temporal dimension. While expanding temporal channels offers a means to alter the length of output predicted frames, a more computationally efficient strategy is sought. Mimicking RNNs provides a solution: RNNs inherently generate long-term forecasts by recycling prior predictions as present inputs. When the desired prediction length $K$ is shorter than the input sequence length $T$, the most recent $K$ timesteps are sliced from the input, adjusting it to $B \times K \times C \times H \times W$. This adjustment ensures that the model's autoregressive predictions align with the intended temporal horizon.

## G  TRANSFERABILITY OF NUWADYNAMICS

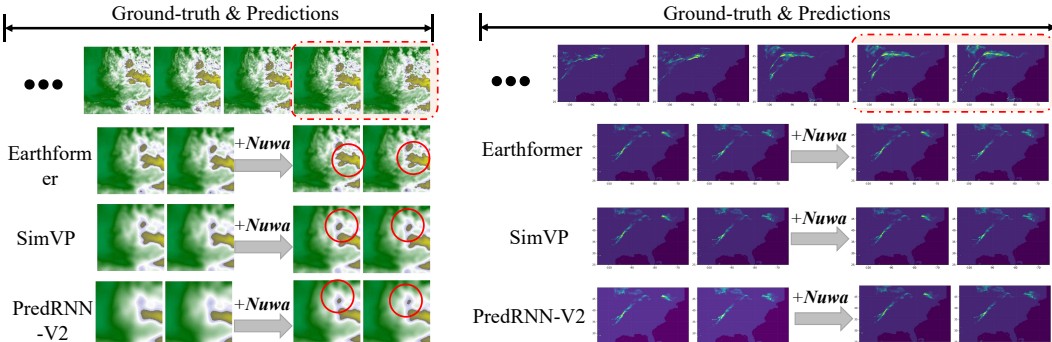

Figure 12: Visualizations of the transfer learning. We only display the last six frames for convenience.

## H  PROOFS OF BACKDOOR ADJUSTMENT IN NUWADYNAMICS

Backdoor adjustment refers to a method used in causal inference to eliminate or control for confounding variables that may affect the relationship between the treatment and the outcome. This is achieved by conditioning on a set of variables (the backdoor criterion) that blocks all backdoor paths from the treatment to the outcome through confounders. By doing so, one can isolate the causal effect of the treatment on the outcome from the biases introduced by confounders. In this work, we employ the backdoor adjustment mechanism to better assist downstream models in perceiving potential test distributions (Yu et al., 2023c;b;a; Liu et al., 2023c).

The do-calculus is a set of three rules introduced by (Pearl et al., 2000) as a part of the causal inference framework. It's a mathematical formalism for reasoning about interventions and causal effects. The do-calculus is utilized for deriving expressions for causal effects in terms of observed distributions, which can be evaluated from data. The rules of do-calculus allow for the manipulation of expressions involving "do" operators, which correspond to interventions in a causal model. As shown in Fig 13, based on the above descriptions, we can apply the following three rules:

- *Rule 1: Insertion/deletion of observations.* $P\left(\mathcal{Y}|do\left(\tilde{C}\right), S\right) = P(\mathcal{Y}|\tilde{C})$ since the environment variable $S$ does not affect the prediction of $\tilde{C}$ to $\mathcal{Y}$, *i.e.,* $\mathcal{Y} \perp S|\tilde{C}$.

- *Rule 2: Action/observation exchange.* $\left(\mathcal{Y}|do\left(\tilde{C}\right), do\left(S\right)\right) = P(\mathcal{Y}|do\left(\tilde{C}\right), S)$ if $\tilde{C}$ is not a descendant of $S$.

- *Rule 3: Rule of Reversal.* $P\left(\tilde{C}|do\left(\mathcal{Y}\right)\right) = P(\tilde{C}|\mathcal{Y})$ if $\tilde{C}$ is not a descendant of $\mathcal{Y}$.

**Backdoor Adjustment.** Based on the above three rules, we showcase the relevance of our algorithm and backdoor adjustment. Our algorithm can be well-understood as a form of backdoor adjustment to enhance the potential data and remove backdoor paths. Uniquely, due to the complexity of environmental variables in the backdoor paths, it significantly increases the training burden. In Section 3.3, we introduced the concept of spatio-temporal (ST) bank to better select influential patches, thereby achieving a trade-off between performance and computational resources.

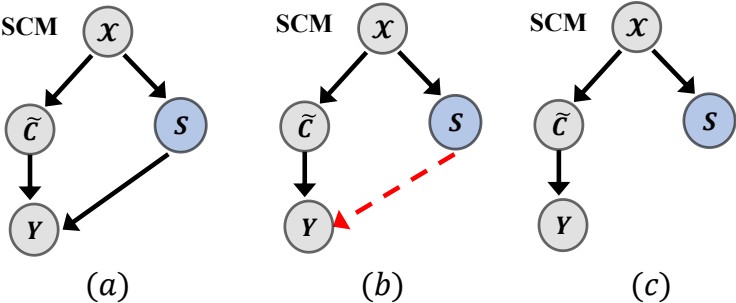

Figure 13: Fig (a) represents the general deep models prediction processes, which consider the environmental parts. Fig (b) illustrates that within the input, there exists an environmental portion $S$. $S$ does not contribute to the model's prediction, which may consequently lead to spurious associations. (c) denotes the model prediction after backdoor adjustment, we can remove spurious correlations by traversing the potential test distributions.

$$
\begin{aligned}
P\left(\mathcal{Y}|do\left(\tilde{C}\right)\right) &= \sum_i^\tau P\left(\mathcal{Y}|do\left(\tilde{C}\right), S = S_i\right) P\left(S = S_i|do\left(\tilde{C}\right)\right) \\
&= \sum_i^\tau P\left(\mathcal{Y}|do\left(\tilde{C}\right), S = S_i\right) P\left(S = S_i\right) \qquad Rule\ 3 \\
&= \sum_i^\tau P\left(\mathcal{Y}|\tilde{C}, S = S_i\right) P\left(S = S_i\right) \qquad Rule\ 1
\end{aligned}
$$

$$(8)$$

## I   DESCRIPTIONS OF ST BANK AND GAUSSIAN SAMPLING

We employ a discretized Gaussian formula to extract data from historical time steps to construct sequences, aiming to better aid the model in creating backups of spatio-temporal prediction data:

$$
\mathcal{G}\left(T, \sigma^2\right) = \frac{1}{\sigma\sqrt{2\pi}} \exp\left(-\frac{(x - T)^2}{2\sigma^2}\right)
$$

$$(9)$$

The aforementioned formula can be further illustrated in Fig 15, we use the current moment $t$ as the mean, and with a variance of $\sigma^2$, apply a Gaussian distribution probability to sample the intervention data in the ST bank. Evidently, in the ST bank, the closer the data is to the current moment, the higher the sampling ratio, and vice versa.

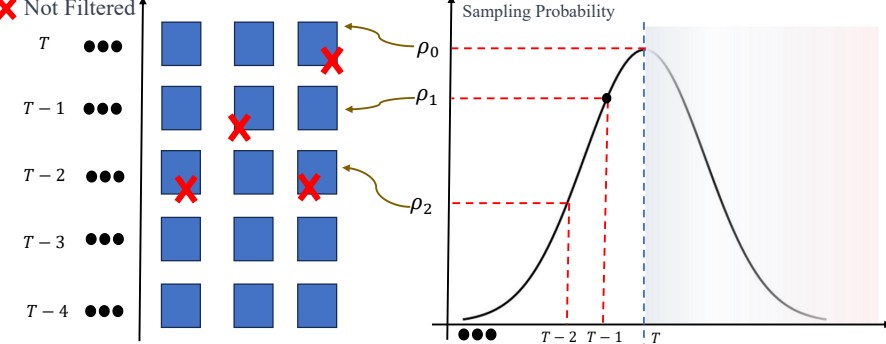

Figure 14: An illustration of discretized Gaussian formula.

## J    ABLATION EXPERIMENTS ON SPATIAL AND TEMPORAL AUGMENTATION

In this section, we meticulously execute three distinct ablations to elucidate the influence of key components within our proposed model. These methodologies are articulated as follows:

1. **Strategy A1**: This method primarily involves the manipulation of only the causal patches at the current time point, deliberately omitting historical patches. This approach is intended to isolate and evaluate the impact of immediate causal effects, devoid of historical influences, thereby providing insight into the temporal immediacy of the model's performance.

2. **Strategy A2**: In this variant, our attention pivots to historical data, but with a significant alteration: all historical patches are accorded equal significance, effectively disregarding the time decay factor. Each time point is uniformly weighted, with a value of 1. This modification is designed to probe the model's sensitivity to temporal variations and to ascertain the importance of differentially weighting historical data based on their temporal proximity.

3. **Strategy A3**: This strategy concentrates on the examination of historical patches specific to the region of interest, while excluding the broader spatial context and other causal patches. These patches are incorporated with a decay factor, with the objective of exploring the localized temporal dynamics and their isolated influence on the model's predictive accuracy.

For clarity and consistency in our analysis, these strategies are systematically designated as **A1**, **A2**, and **A3**. The outcomes of these ablation tests, particularly in terms of MAE, are tabulated. This structured presentation is chosen to enable a lucid comparison and an in-depth understanding of the distinct and collective impacts of these strategies on our model's efficacy. Through this comprehensive ablation study, we aim to unravel the complex inter-dependencies among causal, temporal, and spatial elements in our analytical construct.

|  | TaxiBJ+ | | | KTH | | | RainNet | | | PD | | |
|---|---|---|---|---|---|---|---|---|---|---|---|---|
|  | A1 | A2 | A3 | A1 | A2 | A3 | A1 | A2 | A3 | A1 | A2 | A3 |
| ViT + SimVP | 2.89 | 2.72 | 3.02 | 37.64 | 35.25 | 43.26 | 1.16 | 1.08 | 1.25 | 40.86 | 35.64 | 50.62 |
| SWin + SimVP | 2.72 | 2.61 | 2.94 | 35.56 | 34.12 | 42.64 | 1.05 | 0.99 | 1.16 | 39.34 | 34.16 | 49.13 |
| ViT + PredRNN | 3.81 | 3.54 | 4.28 | 45.42 | 42.06 | 51.25 | 2.55 | 2.49 | 2.62 | 84.36 | 79.92 | 92.86 |
| SWin + PredRNN | 3.72 | 3.41 | 4.15 | 44.32 | 41.17 | 50.14 | 2.45 | 2.48 | 2.53 | 82.32 | 77.43 | 90.95 |

Table 5: Ablation study results showing MAE across different datasets and backbones.

The comparison across strategies indicates that **Strategy A2**, which treats all historical data equally and does not consider time decay, typically offers superior performance. This suggests the predominance of historical data in enhancing predictive accuracy. Conversely, **Strategy A1**, which concentrates solely on the current causal patches, is advantageous for datasets where present data is more indicative of future outcomes. However, **Strategy A3** leads to increased MAE for the KTH and PD datasets, underscoring the significance of spatial context in these scenarios. Collectively, these insights reveal that while historical data is paramount, spatial context is indispensable for datasets with intricate spatial dependencies.

## K    ADDITIONAL RESULTS ON HIGH-RESOLUTION DATASETS

In order to deeply study Nuwa's capability in handling high-resolution datasets (Khojasteh et al., 2022), we chose a cylinder dataset with 768x768 resolution to systematically validate the performance and effectiveness of our algorithm. In order to maintain the consistency of the main paper, we chose ViT, Swin Transformer, Rainformer, Earthformer, as well as ConvLSTM, PredRNN-V2, E3D-LSTM, and SimVP as the backbone networks for our experiments. We ensure that the experimental setup remains consistent with the main part.

The implementation of Nuwa across various deep learning models consistently enhances model accuracy, with notable reductions in both MSE and MAE metrics. The Earthformer model exhibits the most dramatic improvement, dropping from 0.49 to 0.33 in MSE and from 0.43 to 0.32 in MAE.

Table 6: Experimental results on top of the high-resolution cylinder dataset

| Backbone | MSE | | MAE | |
| --- | --- | --- | --- | --- |
| | Ori | +NuWa | Ori | +NuWa |
| ViT | 0.67 | 0.37 | 0.56 | 0.37 |
| SwinT | 0.61 | 0.34 | 0.55 | 0.33 |
| Rainformer | 0.55 | 0.49 | 0.36 | 0.40 |
| Earthformer | 0.49 | 0.33 | 0.43 | 0.32 |
| ConvLSTM | 0.61 | 0.48 | 0.53 | 0.38 |
| PredRNN-V2 | 0.70 | 0.61 | 0.49 | 0.39 |
| E3D-LSTM | 0.54 | 0.31 | 0.43 | 0.31 |
| SimVP | 0.40 | 0.31 | 0.37 | 0.30 |

The experimental results also show the robustness of SimVP, which maintains the lowest MSE and MAE both before and after the enhancement using Nuwa. While the Rainformer model displays an unusual increase in MAE, suggesting a trade-off introduced by NuWa Improvements vary among models, with Earthformer, SimVP, and E3D-LSTM benefiting substantially, illustrating NuWa's variable impact.

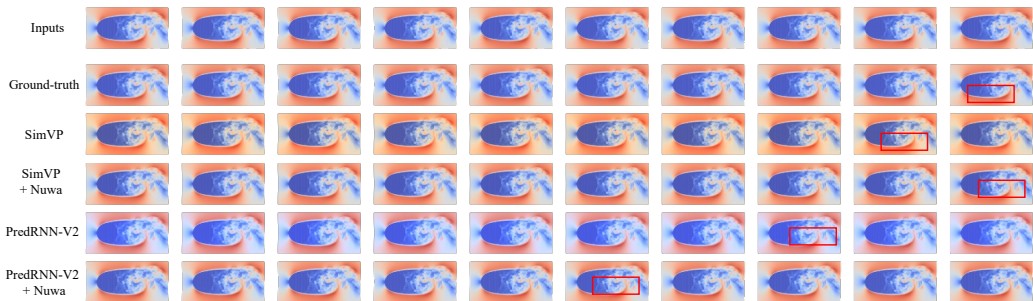

Figure 15: Visualization of spatiotemporal prediction with cylinder dataset. The figure demonstrates that upon incorporating Nuwa, the model achieves greater precision in details, with predictions displaying more accurate textural information.

## L  EVALUATIONS OF CAUSAL DISCOVERY

In light of the lack of a well-defined causal region in traditional datasets, this section elucidates the accuracy of our causal discovery through a real pedestrian movement dataset. We select the Human3.6m for upstream self-supervised tasks and employ attention maps to visualize the areas of importance Ionescu et al. (2013). The results, as depicted in the following figure, demonstrate that our method proficiently correlates pedestrians with causal regions.

As illustrated in the accompanying Figure 16. It is readily apparent that our upstream model proficiently identifies significant areas. These results serve to validate the causal discovery capabilities of our algorithm.

## M  FUTURE WORK

In this paper, we have pioneered the investigation into the integration of causal reasoning with spatio-temporal observable data, systematically formulating a solution that addresses out-of-distribution generalization issues in spatio-temporal contexts. By implementing a Mixup data augmentation technique, we have laid the groundwork for substantial improvements in model performance. Recognizing the potential for further advancements. Our future work will delve into extending the application of our work to a broader spectrum of downstream graph learning tasks, such as graph pruning (Wang et al., 2023b; Xia et al., 2023; Wang et al., 2023c;a), graph sparification (Li et al., 2023; Wu

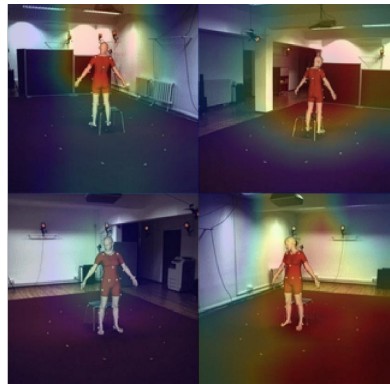 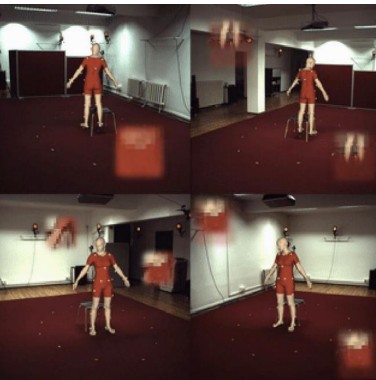

Figure 16: *Left.* Attention map regions on the Human3.6m dataset. *Right.* Augmented samples by Nuwa.

et al., 2023a; Wang et al., 2024; Zhang et al., 2024; Fang et al., 2024a) and graph explainability (Fang et al., 2023b;c; 2022). Moreover, we plan to explore the utility of stable diffusion (Rombach et al., 2022; Zhang et al., 2023) for environmental patch enhancement and leverage Language Model Learning (LLM) (Liu et al., 2023b; Fang et al., 2024b) for more sophisticated data description and send these textual information for guide agumentation. These endeavors aim to refine our model's predictive capabilities and generalizability, thereby contributing to the evolution of robust analytical tools in spatio-temporal data analysis. Interestingly, our method's ability to identify causal components can be further refined and optimized using advanced quantitative metrics. A notable example is the OAR metric (Fang et al., 2023a), which takes the first step to tackle the inherent OOD issues of traditional metrics in deep learning explainability domains. On the other hand, we acknowledge that there is still room for improvement in the efficiency of Nuwa, which can be optimized using the latest pruning paradigms. For instance, RGLT (Wang et al., 2023a) achieves joint pruning of data and networks while maintaining generalization and robustness through causal pruning theory.

|  | TaxiBJ+ | | | KTH | | | RainNet | | | PD | | |
|---|---|---|---|---|---|---|---|---|---|---|---|---|
|  | A1 | A2 | A3 | A1 | A2 | A3 | A1 | A2 | A3 | A1 | A2 | A3 |
| ViT + SimVP | 2.89 | 2.72 | 3.02 | 37.64 | 35.25 | 43.26 | 1.16 | 1.08 | 1.25 | 40.86 | 35.64 | 50.62 |
| SWin + SimVP | 2.72 | 2.61 | 2.94 | 35.56 | 34.12 | 42.64 | 1.05 | 0.99 | 1.16 | 39.34 | 34.16 | 49.13 |
| ViT + PredRNN | 3.81 | 3.54 | 4.28 | 45.42 | 42.06 | 51.25 | 2.55 | 2.49 | 2.62 | 84.36 | 79.92 | 92.86 |
| SWin + PredRNN | 3.72 | 3.41 | 4.15 | 44.32 | 41.17 | 50.14 | 2.45 | 2.48 | 2.53 | 82.32 | 77.43 | 90.95 |

Table 7: Ablation study results showing MAE across different datasets and backbones.

