# OpenReview forum: "NuwaDynamics: Discovering and Updating in Causal Spatio-Temporal Modeling"
_ICLR.cc/2024/Conference — ICLR 2024 spotlight_

### Official Review · Reviewer_TYFb · 2023-10-13

**Soundness:** 4 excellent
**Presentation:** 4 excellent
**Contribution:** 4 excellent
**Rating:** 10
**Confidence:** 4

**Summary:**

The paper delves into the spatio-temporal challenges inherent in earth sciences, encompassing areas like meteorological forecasting and urban computing. Given the hurdles of data sparsity and imbalance, the authors put forth a novel causal prediction framework dubbed "NuwaDynamics". Operating through two phases - "Discovery" and "Update" - this paradigm pinpoints causal regions within datasets and endows the model with causal reasoning prowess. The authors also substantiate their methodology with proofs drawn from causal theory and employ a myriad of experimental results to demonstrate its efficacy across various spatio-temporal benchmark tests, especially in tasks like extreme weather forecasting and long-term temporal predictions. In a nod to fostering collaboration and transparency, the code from this research has been shared with the public, marking a valuable contribution to the open-source arena.

**Strengths:**

1. The paper is well-written and exhibits a commendable layout; the motivation is intuitive and clear.

2. It is interesting to employ ViT and Mixup in spatiotemporal data for causal environmental discovery and augmentation, and the work can potentially expand the model's latent observational space in many scarce scenarios.

3. Extensive experiments demonstrate the effectiveness of the framework. The authors introduced architectures tailored for both ViT and non-ViT, while also including control experiments across frameworks such as RNN and CNN. Comparisons across multiple mainstream tasks, including transfer learning and super-resolution prediction, further demonstrates its capabilities in tasks like extreme event prediction.

4. Detailed theoretical validation based on causal manipulation is provided to demonstrate the feasibility of the framework. Interestingly, in real-world scenarios, the author doesn't explore all possible environments but instead introduces partial perturbations, which conveniently addresses the computational constraints and limited data availability in practice.

**Weaknesses:**

1. I am concerned about whether there are clear guiding principles for environmental augmentation. Especially when it comes to the mixup technique, is it possible to automatically adjust the fusion hyperparameters of each environmental patch based on a specific dataset? If this cannot be achieved at the moment, I hope the authors will offer potential solutions in their section of future work .

2. I also have a keen interest in environmental perturbation methods. In fact, environmental perturbation can be considered a form of data augmentation. I wonder, are there any generative methods in computer vision (CV) that can be integrated into this? If the answer is affirmative, I hope the authors delve deeper into this in the "Related Work" and "Model" sections.

3. The experimental section of the paper is rich in content, and I am particularly interested in spatio-temporal system modeling. Current research literature mainly focuses on datasets with lower resolution, for example, roughly 256x256 in size. I hope the authors can consider more experiments to explore whether these methods remain competitive on high-resolution datasets. If the authors can achieve superior results on higher-resolution datasets, I might consider further raising my score.

**Questions:**

See weaknesses.

---

> ### Author Response · Authors · 2023-11-12
> **Response to Reviewer TYFb _ 1**
>
> Thank you for your careful and detailed review of our manuscript and for your highly appreciative evaluation of our work. Based on the your questions and recommendations, we have made many revisions that significantly improved the clarity of the paper. The new, revised paper has been uploaded to OpenReview.
>
> #### Q1: whether there are clear guiding principles for environmental augmentation.
>
> We establish a clear set of guidelines for their determination. Specifically, we control the intervention weight coefficients for environmental patches based on the weights in the attention map, and we also consider the impact of temporal decay on these coefficients, integrating both aspects into our environmental enhancement (see Equations 3-5). We have systematically deconstructed our data augmentation into two aspects to address your question more thoroughly:
>
> ###### Spatial perspective
>
> we employ the Mixup technique because the state of a given region  is influenced by surrounding areas, and also consider the importance of surrounding areas to assign weights to different regions. The Mixup technique is adept at extracting pertinent information from causally related neighboring regions and efficiently constructs stable regional expressions.
>
> ###### Temporal perspective
>
> Inspired by studies on point processes [1-2], we postulate that the influence over time is progressively diminishing. We believe that spatio-temporal states closer to the current moment have a greater impact on future predictions. Accordingly, we have introduced a temporal decay enhancement method, which systematically reduces the number of enhancements as they approach the future moment in question.
>
> In our analysis, we implement three distinct ablation methods. For the first strategy, **A1**, we mix up only the causal patches at the current time point, not considering historical patches. For the second strategy, **A2**, we do not account for the time decay of historical patches, instead giving equal weight to all historical data (assigning a weight of 1 to each time point). As for the third strategy, **A3**, we focus on the historical patches of the region itself without considering the spatial context and causal patches, adding them with decay. We organize these ablation strategies as **A1**, **A2**, and **A3** respectively, and present the results (MAE) in a structured table format to facilitate a clear understanding of their impacts.
>
> | Model Combination | TaxiBJ+ A1 | TaxiBJ+ A2 | TaxiBJ+ A3 | KTH A1 | KTH A2 | KTH A3 | RainNet A1 | RainNet A2 | RainNet A3 | PD A1 | PD A2 | PD A3 |
> | ----------------- | ---------- | ---------- | ---------- | ------ | ------ | ------ | ---------- | ---------- | ---------- | ----- | ----- | ----- |
> | ViT + SimVP       | 2.89       | 2.72       | 3.02       | 37.64  | 35.25  | 43.26  | 1.16       | 1.08       | 1.25       | 40.86 | 35.64 | 50.62 |
> | SWin + SimVP      | 2.72       | 2.61       | 2.94       | 35.36  | 34.12  | 42.64  | 1.05       | 0.99       | 1.16       | 39.34 | 34.16 | 49.13 |
> | ViT + PredRNN     | 3.81       | 3.54       | 4.28       | 42.65  | 42.06  | 51.25  | 2.55       | 2.49       | 2.62       | 84.36 | 79.92 | 92.86 |
> | SWin + PredRNN    | 3.72       | 3.41       | 4.15       | 44.32  | 41.17  | 50.14  | 2.45       | 2.48       | 2.53       | 82.32 | 77.43 | 90.95 |
>
> The ablation study shows that the performance significantly drops for **Strategy A3**, which ignores the spatial context and does not utilize causal patches for mixup. We suspect the potential reason is: if the region itself is an unimportant environmental patch, then applying a historical decay weighting to these environmental patches is meaningless. This underscores the importance of our nuwa causal reasoning. When we consider **Strategy A1**, which takes into account causal patches, there is a clear improvement in performance compared to **Strategy A3**. Finally, **Strategy A2**—which considers the impact of historical patches but assigns equal weight to all historical data without time decay—outperforms the other strategies, indicating that historical data plays a crucial role in predictive accuracy. Overall, none of these strategies perform as well as nuwa, which takes into account both the impact of spatial causal patches and the effect of time decay, proving that our approach is comprehensive.
>
> For ease of understanding, we have incorporated additional ablation studies into the original manuscript to elucidate the robustness and effectiveness of our proposed methods. Furthermore, we have expanded the section on **Future Work** to include potential augmentation strategies.
>
>
>
> [1] Zuo, Simiao, et al. "Transformer hawkes process." *International conference on machine learning*. PMLR, 2020.
>
> [2] Mei, Hongyuan, and Jason M. Eisner. "The neural hawkes process: A neurally self-modulating multivariate point process." *Advances in neural information processing systems* 30 (2017).

---

> ### Author Response · Authors · 2023-11-12
> **Response to Reviewer TYFb _ 2**
>
> #### Q2: Are there any generative methods in computer vision (CV) that can be integrated into data augmentation?
>
> Yes,  We believe that works related to stable diffusion and Generative Adversarial Networks (GANs), can effectively enhance data augmentation [3] [4]. We have integrated these concepts into the Related Work and Methodology sections of our paper to more comprehensively articulate the efficacy and efficiency of our methods. This integration not only underscores the relevance of stable diffusion and GANs in the context of our research but also highlights their potential in improving the robustness and performance of data-driven models.
>
>
>
> [3] Creswell, Antonia, et al. "Generative adversarial networks: An overview." *IEEE signal processing magazine* 35.1 (2018): 53-65.
>
> [4] Croitoru, Florinel-Alin, et al. "Diffusion models in vision: A survey." *IEEE Transactions on Pattern Analysis and Machine Intelligence* (2023).
>
> ####  Q3: Conduct experiments to explore whether these methods remain competitive on high-resolution datasets.
>
> In response to the your suggestion for a more comprehensive exploration of Nuwa's capabilities with high-resolution datasets, we have selected a 768x768 resolution cylinder dataset [5] to systematically validate the effectiveness of our algorithm. This dataset choice specifically addresses the performance of Nuwa in handling high-resolution scenarios, allowing us to demonstrate its robustness and accuracy in more demanding contexts.
>
>
> | Backbone    | MSE Ori | MSE +NuWa | MAE Ori | MAE +NuWa |
> | ----------- | ------- | --------- | ------- | --------- |
> | ViT         | 0.67    | 0.37      | 0.56    | 0.37      |
> | SwinT       | 0.61    | 0.34      | 0.55    | 0.33      |
> | Rainformer  | 0.55    | 0.49      | 0.36    | 0.40      |
> | Earthformer | 0.49    | 0.33      | 0.43    | 0.32      |
> | ConvLSTM    | 0.61    | 0.48      | 0.53    | 0.38      |
> | PredRNN-V2  | 0.70    | 0.61      | 0.49    | 0.39      |
> | E3D-LSTM    | 0.54    | 0.31      | 0.43    | 0.31      |
> | SimVP       | 0.40    | 0.31      | 0.37    | 0.30      |
>
>
>
>
>
> [5] Khojasteh, Ali Rahimi, et al. "Lagrangian and Eulerian dataset of the wake downstream of a smooth cylinder at a Reynolds number equal to 3900." *Data in brief* 40 (2022): 107725.

---

> > ### Comment · Reviewer_TYFb · 2023-11-13
> >
> > Dear Authors,
> >
> > Many thanks for your detailed and prompt rebuttal. I really learned much from it about the generative methods for data augmentation. Besides that, my concerns are fully addressed. I believe that your paper makes a substantial contribution to our field and will be of great interest to the research community, and I would like to raise my score to strongly support the recommendation for acceptance.
> >
> > Best,
> > Reviewer TYFb

---

### Official Review · Reviewer_oBVm · 2023-10-26

**Soundness:** 4 excellent
**Presentation:** 3 good
**Contribution:** 4 excellent
**Rating:** 8
**Confidence:** 5

**Summary:**

This paper explores the application of deep learning to dynamical systems and proposes a new philosophical framework called "NuwaDynamics", which aims to mine data for causal patterns to enhance the explanatory and generalization capabilities of models.  The approach is divided into two phases: Discovery, which identifies the causal components of the data, and Update, which applies the causal model to downstream tasks.  In addition, NuwaDynamics enhances the model's performance in sparse and extreme situations.  Experimental results in the article show that the framework achieves excellent results in several benchmark tests.

**Strengths:**

(1) A conceptually and technically highly innovative paper which seamlessly integrates causal theory with spatio-temporal data mining. The integration of causal theory's environment discovery and augmentation into the popular vision transformer's attention map is ingenious. Coupled with a straightforward mixup augmentation, it achieves a very refined alignment. From my perspective, NuwaDynamics has the potential to revolutionize how models recognize causality in ST data mining realm. This may provide a new approach to solving such problems (data scarcity and the high cost of deploying sensors).

(2) The paper reads very well, and hardly has any errors or inconsistencies. Information is provided at the right point, is complete and accurate. The split between main paper and supplementary material is good. The narrative and exposition flow well. Meanwhile, the paper is well-structured, I appreciate its visual aids.

(3) Providing a causal intervention perspective for data augmentation seems interesting. In terms of practical implementation, utilizing attention scores without introducing additional complex designs seems both simple and effective. Additionally, I believe the backdoor adjustment mechanism proposed by the authors serves as a robust theoretical foundation. This contributes significantly to enhancing generalization, especially in out-of-distribution scenarios.

(4) Baselines are strong, relevant, discussed well and evaluated fairly. A large number of analytic experiments complete the main findings. Meanwhile, experimental setups are presented accurately and consistently, at an adequate level of detail.

**Weaknesses:**

(1) The paper does not seem to make it clear exactly how the process of obtaining the importance rankings in the patches is done, and it is hoped that the authors will add that detail.

(2) In referring to the fact that DL methods often forgo an explicit understanding of the rules of physics, the practical implications and potential risks of this sacrifice are not explained in detail. More discussion or explanation might have given the reader a better understanding of the consequences of this choice.

**Questions:**

(1) Despite the extensive experiments presented in this paper, I'm still curious about the effectiveness of this method on certain spatio-temporal datasets, such as [1-2]. These real pedestrian movement datasets could offer a more convincing validation of the model's efficacy.  However, an intriguing question arises: given that Nuwa's environment is populated with causal patches, is its performance reliable on pedestrian movement datasets that have clear divisions between causal and non-causal segments?


[1] Pedestrian detection: A benchmark  **Conference on Computer Vision and Pattern Recognition**
[2] Human3. 6m: Large scale datasets and predictive methods for 3d human sensing in natural environments **IEEE transactions on pattern analysis and machine intelligence**

---

> ### Author Response · Authors · 2023-11-12
> **Response to reviewer oBVm**
>
> We are deeply grateful for the comprehensive and insightful review you have provided for our manuscript, as well as for your positive feedback on our research efforts. Following your astute observations and valuable guidance, we have incorporated additional experimental evaluations with novel datasets, aiming to thoroughly respond to your queries. With these substantial improvements, we have meticulously revised the manuscript and have subsequently re-uploaded it to OpenReview for additional assessment and feedback.
>
> #### W1: The paper does not seem to make it clear exactly how the process of obtaining the importance rankings in the patches is done, and it is hoped that the authors will add that detail.
>
> In our paper, we identify key tokens within the attention map by summing and ranking the columns, pinpointing the tokens corresponding to the largest values, and designating those with the smallest values as the environmental part.  Due to variations across datasets, the proportion of the environmental part slightly differs.  To aid reader comprehension, we have incorporated new descriptions within the manuscript that clarify these dataset-specific differences.
>
>
>
> #### W2: More discussion or explanation might have given the reader a better understanding of the consequences of using data-driven approach rather than PINN.
>
> Data-driven approaches are favored in various fields due to their unique advantages. This approach can effectively deal with systems of high complexity and autonomously mine out new correlations from data, which greatly improves the efficiency of computation. More importantly, it reduces the reliance on deep expertise and makes the modeling process easier [1] [2] [3]. In environments where data is abundant and needs are evolving, data-driven modeling demonstrates its strong adaptability and swift predictive capabilities. Moreover, since not all contexts can be explained by physical rules, the scalability and generalizability of data-driven models make them ideal for building pervasive models.
>
>
>
> [1] Cai, Shengze, et al. "Physics-informed neural networks (PINNs) for fluid mechanics: A review." *Acta Mechanica Sinica* 37.12 (2021): 1727-1738.
>
> [2] Krishnapriyan, Aditi, et al. "Characterizing possible failure modes in physics-informed neural networks." *Advances in Neural Information Processing Systems* 34 (2021): 26548-26560.
>
> [3] Cuomo, Salvatore, et al. "Scientific machine learning through physics–informed neural networks: Where we are and what’s next." *Journal of Scientific Computing* 92.3 (2022): 88.
>
> #### Q1: Add new experiments on real pedestrian movement datasets.
>
>
>
> We conduct experiments on the Human3.6m dataset, which features real pedestrian movement, and utilize attention maps to visualize the high-scoring regions. In these maps, we define pedestrians as the causal components. Our model adeptly identifies these causal regions. To present our findings more effectively, we include a new section in the Appendix K (**Evaluations of causal discovery**) dedicated to the validation of causal regions.

---

> > ### Comment · Reviewer_oBVm · 2023-11-22
> >
> > my concerns are well addressed.
> > Thanks.

---

### Official Review · Reviewer_zp2y · 2023-10-30

**Soundness:** 2 fair
**Presentation:** 3 good
**Contribution:** 3 good
**Rating:** 6
**Confidence:** 3

**Summary:**

This paper established a causal concept for spatio-temporal predictions. In particular, it firstly identifies causal regions at the Discovery step. It then augments non-causal patches at the Update stage. By doing so, the model is able to see a broader potential distribution of data, achieving improved downstream performance. Extensive experiments show that the proposed model can benefit many existing frameworks on various tasks.

**Strengths:**

**Originality:** This paper established a causal concept for spatio-temporal predictions and introduced a novel philosophical framework. Its causal perspective is interesting and original. It is also addressing an important challenge faced by AI community which is the interpretability and generalizability.

**Presentation:** This paper is well written. The flow of the paper is smooth and easy to follow. Figure 1 and Figure 2 are very helpful to illustrate the central idea of the paper.

**Experiments:** the experiments are extensive, covering a wide range of backbones and tasks. The experimental analysis is well organized with key questions and itemized observations.

**Weaknesses:**

**The experiments are insufficient.** The performed experiments are focused on the comparison between the performance of an original backbone and the performance of adding NuWa. There is a lack of comparison to SOTA performance on the evaluated tasks. Besides, it is also important to compare to other data-augmentation methods (even though they are not using causality), which is also missing in the paper.

Besides, there is a lack of evaluation on the causal discovery accuracy. Causal discovery always requires a large amount of data to ensure accurate discovery of causal relationships, and it can suffer from highly imbalanced data. How does your model address this issue as you are particularly interested in such regime?

**Questions:**

Please see the weakness section for my major concerns. In addition,

1.	How many augmented samples are required to achieve reasonable performance? Does the downstream performance sensitive to the number of augmented samples?
2.	What if we directly perform data enhancement using the attention map?
3.	Can you show the visualization of the identified causal regions and the augmented samples?

---

> ### Author Response · Authors · 2023-11-13
> **Response to Reviewer zp2y_1**
>
> We express our sincere thanks for the detailed and thoughtful review of our manuscript and for the encouraging appraisal of our work. In response to your perceptive questions and constructive recommendations, we have conducted extensive ablation studies and added experimental evaluations using new datasets to comprehensively address your inquiries.
>
> #### W1: Add augmentation experiments.
>
> In the field of spatio-temporal prediction, we observe that conventional data augmentation methods often lead to a decrease in predictive performance. For instance, operations like flipping or rotating images alter the physical dynamics of spatiotemporal data, thereby affecting the model's understanding of temporal continuity and dynamic changes. In contrast, data augmentation methods based on causal relationships show a significantly different effect. These methods, by deeply understanding the causal mechanisms behind the data, can effectively enhance the predictive capability of the model, thereby improving the accuracy and robustness of spatiotemporal predictions. This underscores the importance of selecting appropriate data augmentation strategies in the analysis and prediction of spatiotemporal data. **B1** is ori+flip, **B2** is ori+rotate, **B3** is ori+VQ-VAE (which can be regarded as currently popular generative CV model). From the table, we can see that traditional data augmentation methods result in performance that is even worse than the original backbone. However, using VQ-VAE for data augmentation indeed improves the performance, but not as effectively as our proposed causality-based augmentation.
>
> | Model Combination | TaxiBJ+ B1 | TaxiBJ+ B2 | TaxiBJ+ B3 | KTH B1 | KTH B2 | KTH B3 | RainNet B1 | RainNet B2 | RainNet B3 | PD B1 | PD B2 | PD B3 |
> | ----------------- | ---------- | ---------- | ---------- | ------ | ------ | ------ | ---------- | ---------- | ---------- | :---: | ----- | ----- |
> | ViT+SimVP         | 4.28       | 4.14       | 3.01       | 48.64  | 47.15  | 39.25  | 2.01       | 2.03       | 1.25       | 2.76  | 2.72  | 1.85  |
> | SwinT+SimVP       | 4.25       | 4.09       | 2.94       | 46.02  | 45.62  | 37.16  | 1.95       | 1.91       | 1.19       | 2.71  | 2.69  | 1.76  |
> | ViT+PredRNN       | 5.53       | 5.45       | 3.86       | 59.06  | 59.01  | 46.31  | 3.79       | 3.81       | 2.62       | 5.01  | 4.99  | 4.26  |
> | SwinT+PredRNN     | 5.49       | 5.41       | 3.72       | 58.45  | 58.62  | 44.57  | 3.71       | 3.69       | 2.54       | 4.96  | 4.91  | 4.19  |
>
> We also conduct ablations from temporal augmentation and spatial augmentation two perspective to address your question more thoroughly: In our analysis, we implement three distinct ablation methods. For the first strategy, **A1**, we mix up only the causal patches at the current time point, not considering historical patches. For the second strategy, **A2**, we do not account for the time decay of historical patches, instead giving equal weight to all historical data (assigning a weight of 1 to each time point). As for the third strategy, **A3**, we focus on the historical patches of the region itself without considering the spatial context and causal patches, adding them with decay. We organize these ablation strategies as **A1**, **A2**, and **A3** respectively, and present the results (MAE) in a structured table format to facilitate a clear understanding of their impacts.
>
> | Model Combination | TaxiBJ+ A1 | TaxiBJ+ A2 | TaxiBJ+ A3 | KTH A1 | KTH A2 | KTH A3 | RainNet A1 | RainNet A2 | RainNet A3 | PD A1 | PD A2 | PD A3 |
> | -------- | ------ | ------ | -------- | ------ | ------ | ------ | ---------- | ---------- | ---------- | ----- | ----- | ----- |
> | ViT + SimVP       | 2.89       | 2.72       | 3.02       | 37.64  | 35.25  | 43.26  | 1.16       | 1.08       | 1.25       | 40.86 | 35.64 | 50.62 |
> | SWin + SimVP      | 2.72       | 2.61       | 2.94       | 36.56  | 34.12  | 42.64  | 1.05       | 0.99       | 1.16       | 39.34 | 34.16 | 49.13 |
> | ViT + PredRNN     | 3.81       | 3.54       | 4.28       | 45.42  | 42.06  | 51.25  | 2.55       | 2.49       | 2.62       | 84.36 | 79.92 | 92.86 |
> | SWin + PredRNN    | 3.72       | 3.41       | 4.15       | 44.32  | 41.17  | 50.14  | 2.45       | 2.48       | 2.53       | 82.32 | 77.43 | 90.95 |
>
> For the first strategy, **A1**, we mix up only the causal patches at the current time point, not considering historical patches. For the second strategy, **A2**, we do not account for the time decay of historical patches, instead giving equal weight to all historical data (assigning a weight of 1 to each time point). As for the third strategy, **A3**, we focus on the historical patches of the region itself without considering the spatial context and causal patches, adding them with decay. We organize these ablation strategies as **A1**, **A2**, and **A3** respectively, and present the results (MAE) in a structured table format to facilitate a clear understanding of their impacts.

---

> ### Author Response · Authors · 2023-11-13
> **Response to Reviewer zp2y_2**
>
> The ablation study shows that the performance significantly drops for **Strategy A3**, which ignores the spatial context and does not utilize causal patches for mixup. We suspect the potential reason is: if the region itself is an unimportant environmental patch, then applying a historical decay weighting to these environmental patches is meaningless. This underscores the importance of our nuwa causal reasoning. When we consider **Strategy A1**, which takes into account causal patches, there is a clear improvement in performance compared to **Strategy A3**. Finally, **Strategy A2**—which considers the impact of historical patches but assigns equal weight to all historical data without time decay—outperforms the other strategies, indicating that historical data plays a crucial role in predictive accuracy. Overall, none of these strategies perform as well as nuwa, which takes into account both the impact of spatial causal patches and the effect of time decay, proving that our approach is comprehensive.
>
> For ease of understanding, we have incorporated additional ablation studies into the original manuscript to elucidate the robustness and effectiveness of our proposed methods. Furthermore, we have expanded the section on **Future Work** to include potential augmentation strategies.
>
> #### W2 & Q3: Showcase the causal discovery accuracy.
>
> Given the absence of explicit definitions for causal and non-causal areas within the dataset, we conduct experiments on the **Human3.6m dataset**, a collection of real pedestrian movement data. In this context, pedestrians can be construed as regions with causal attributes, while the remaining areas can be considered as environmental. We utilize attention maps from upstream tasks to visualize significant regions, thereby highlighting our capability for causal discovery (see response to reviewer **oBVm**).
>
> In these maps, we define pedestrians as the causal components. Our model adeptly identifies these causal regions. To present our findings more effectively, we include a new section in the Appendix K (**Evaluations of causal discovery**) dedicated to the validation of causal regions and augmentated samples.
>
> **We further claim that the assumption of causality is highly universal; it is not confined to datasets with explicit causal definitions. In addressing out-of-distribution scenarios, many studies [1-5] employ causal discovery methods to circumvent spurious correlations that may cause interference.**
>
> [1] Invariant risk minimization.
>
> [2]  The risks of invariant risk minimization.
>
> [3] Does invariant risk minimization capture invariance?.
>
> [4] Discovering invariant rationales for graph neural networks.
>
> [5] Causal attention for interpretable and generalizable graph classification.
>
>
>  Q1: How many augmented samples are required to achieve reasonable performance?
>
> Firstly, our augmentation ratio is determined through Gaussian sampling, which allows us to enhance the dataset to a specific proportion in a controlled manner. We have observed that different datasets exhibit slight variations in sensitivity to the augmentation ratio; however, the overall impact remains minimal. We have selected TaxiBJ+, KTH, and FireSys datasets for testing to demonstrate the sensitivity of our model to the 'number of augmented samples' parameter. Additionally, we have discovered that varying the augmentation ratios does not significantly affect the model's predictive performance, which further demonstrates the model's stability.
>
> | number of augmented samples | TaxiBJ+ |  KTH  | FireSys |
> | ----- | :-----: | :---: | :-----: |
> | 3                           |  2.79   | 34.86 |  1.71   |
> | 4                           |  2.61   | 34.12 |  1.60   |
> | 5                           |  2.56   | 33.98 |  1.54   |
> | 6                           |  2.54   | 33.95 |  1.51   |
> | 7                           |  2.53   | 33.93 |  1.50   |
>
> The results indicate that after applying Nuwa, there is a improvement in predictive performance across the board.  However, taking into account the trade-off between computational overhead and performance, we have opted for a Gaussian sampling frequency with a smaller variance.  This choice is aimed at better highlighting the efficiency of the model.
>
>
>  Q2: What if we directly perform data enhancement using the attention map?
>
> In fact, we do use the attention map to identify environmental patches for data augmentation. We calculate the importance score of each patch by summing up the attention map column-wise. Based on these importance scores, we select environment patches in different proportions for different datasets. Then, we perform causal mixup enhancement on the environment patches, taking into account both their importance weights and historical patches.
>
> We are currently supplementing our manuscript with additional experimental results and detailed descriptions. We are also very open to and welcome any new questions you may have.

---

> > ### Comment · Reviewer_zp2y · 2023-11-21
> > **Responses to Authors**
> >
> > Dear Authors,
> >
> > Thank you for the extensive ablation studies and experimental evaluations. My concerns are addressed. I find this work interesting and see potential benefits for our research community. I would love to increase my rating to support the recommendation for acceptance.
> >
> > Best,
> > Reviewer zp2y

---

> > > ### Author Response · Authors · 2023-11-22
> > > **Thank you for your constructive feedback!**
> > >
> > > We are gratified to hear that your concerns have been addressed and that you find our work interesting and potentially beneficial to the research community. Your willingness to increase the rating and support the recommendation for acceptance is greatly appreciated. We remain committed to contributing valuable insights to our field and are thankful for your constructive feedback throughout the review process

---

### Official Review · Reviewer_PkBe · 2023-10-31

**Soundness:** 2 fair
**Presentation:** 3 good
**Contribution:** 2 fair
**Rating:** 6
**Confidence:** 5

**Summary:**

This paper proposes a novel causal framework called NuwaDynamics for spatio-temporal (ST) prediction tasks. The key idea is to leverage self-supervised reconstruction tasks to identify causal regions in the input data, and perform interventions on non-causal regions to expose the model to a broader distribution of data. The framework consists of two main stages: 1) Causal patch discovery, where a Vision Transformer is trained on a reconstruction task and attention maps are used to localize causal patches. 2) Causal model update, where environmental patches are modified through interventions and used to train the downstream model. Experiments on 6 benchmarks demonstrate improved performance over strong baselines.

**Strengths:**

1. This paper is well-organized and clearly written.
2. Introducing causality concepts into spatio-temporal modeling is novel.
3. The motivation is clear and reasonable.
4. Visualizations demonstrate improved detail and extreme weather perception.
5. The experiments across multiple datasets are thorough.

**Weaknesses:**

1. It seems that the computational complexity is high due to generating many intervened sequences.
2. Lack of ablation studies to validate design choices.
3. Except for the KTH dataset, the performance gains are relatively low.
4. Lacks of the comparison on computational complexity.

**Questions:**

1. What are the limitations of current attention map methods for identifying causal regions?
2. How sensitive is the approach to the choice of interventions?

---

> ### Author Response · Authors · 2023-11-14
> **Response to Reviewer PkBe _1**
>
> We express our sincere thanks for the detailed and thoughtful review of our manuscript and for the encouraging appraisal of our work. In response to your perceptive questions and constructive recommendations, we have conducted extensive ablation studies and added experimental evaluations using new datasets to comprehensively address your inquiries.
>
> #### W1 & W4 : Computational complexity
>
> Thank you for raising such insightful questions. To better elucidate the time complexity of our approach, we will systematically discuss the algorithmic complexity from two perspectives:
>
> 1. Both upstream and downstream processes are based on the transformer framework and the only difference in the data between the two is the introduction of new temporal sequences after causal interventions. Consequently, the parameters from the upstream can be directly reused by the downstream model.  Here, the parameters from the upstream transformer-based model are also reused, which leads to faster convergence. Therefore, **the computational burden does not increase exponentially but only adds an affordable computational load**.
>
> 2. When the upstream and downstream are inconsistent, upstream self-supervised task aims to identify causal regions. Once the upstream model training is complete, **it can be reused**, avoiding the burden of repetitive training [1-3]. In the downstream, although the number of sequences has increased, **since most sequences are essentially similar in information, the convergence speed does not lead to proportional cost increases**.
>
> We have observed that all datasets reach convergence within 100 epochs on an A100-PCIE-40GB. Since the **upstream training is reusable**, it only needs to be trained once and can be used repeatedly. We therefore only present the training time per epoch for the downstream (upstream is ViT). Here, we have chosen **SimVP as the backbone due to its superior efficiency characteristics compared to others like PredRNN-V2.**
>
> |  | w/o Nuwa, upstream | downstream   | Nuwa, upstream | Nuwa,downstream |
> | ---- | --- | ----- | ---- | ---- |
> | SEVIR   | --| 4 m/epoch  | 6m/epoch     | 8.5m/epoch    |
> | TaxiBJ+ | --| 1.5m/epoch | 2m/epoch     | 4m/epoch      |
> | FireSys | --| 2m/epoch   | 3m/epoch     | 4.5m/epoch    |
>
> Here, we control our sequence augmentation ratio to be nearly fivefold. We find that the efficiency does not increase by five times but only about two to three times. This further confirms that our algorithm does not cause a significant computational burden. We have added an analysis of the **algorithm's complexity to the appendix** to aid readers in better understanding the efficacy and efficiency of our algorithm. Additionally, we have elucidated in the model the rationale behind adopting Gaussian sampling to reduce the computational load during inference.
>
> [1] Videogpt: Video generation using vq-vae and transformers.
>
> [2] Generating diverse high-fidelity images with vq-vae-2.
>
> [3]Deep learning for physical processes: Incorporating prior scientific knowledge.
>
> #### W2: Lack of ablation studies to validate design choices.
>
> Thank you for your suggestion. Indeed, based on your advice, we have included additional ablation experiments, which encompass a variety of **augmentation methods** and **intervention sensitivity experiments** (**see response to Q2**). We have also added an ablation study on the model's augmentation ratios, as indicated in the following table. Moreover, we have incorporated visualizations of the **causal discovery regions** and the **augmented samples** in the revised version (**Appendix K**) of our manuscript, furthering the systematic address of your queries.
>
> Firstly, our augmentation ratio is determined through Gaussian sampling, which allows us to enhance the dataset to a specific proportion in a controlled manner. We have observed that different datasets exhibit slight variations in sensitivity to the augmentation ratio; however, the overall impact remains minimal. We have selected TaxiBJ+, KTH, and FireSys datasets for testing to demonstrate the sensitivity of our model to the 'number of augmented samples' parameter. Additionally, we have discovered that varying the augmentation ratios does not significantly affect the model's predictive performance, which further demonstrates the model's stability.
>
> | number of augmented samples | TaxiBJ+ |  KTH  | FireSys |
> | --- | :--: | :---: | :--: |
> | 3 |  2.79   | 34.86 |1.71  |
> | 4  |  2.61   | 34.12 |1.60 |
> | 5 |  2.56   | 33.98 |1.54 |
> | 6  |  2.54   | 33.95 |1.51 |
> | 7 |  2.53   | 33.93 |1.50 |
>
> The results indicate that after applying Nuwa, there is a improvement in predictive performance across the board.  However, taking into account the trade-off between computational overhead and performance, we have opted for a Gaussian sampling frequency with a smaller variance.  This choice is aimed at better highlighting the efficiency of the model.

---

> > ### Author Response · Authors · 2023-11-14
> > **Response to Reviewer PkBe _3**
> >
> > #### Q2: How sensitive is the approach to the choice of interventions?
> >
> > Our method utilizes mixup as a spatio-temporal intervention augmentation strategy, encompassing both temporal and spatial augmentations. To assess our algorithm's sensitivity to interventions, we have conducted ablation studies for both temporal and spatial (**see response to zp2y**). Furthermore, we have performed ablations on data augmentation methods, comparing generic spatio-temporal sequence enhancements with Nuwa's causality-based regional enhancements. This comparison serves to underscore the rationale behind our algorithm.
> >
> > **B1** is ori+flip, **B2** is ori+rotate, **B3** is ori+VQ-VAE (which can be regarded as currently popular generative CV model). From the table, we can see that traditional data augmentation methods result in performance that is even worse than the original backbone. However, using VQ-VAE for data augmentation indeed improves the performance, but not as effectively as our proposed causality-based augmentation.
> >
> > | Model Combination | TaxiBJ+ B1 | TaxiBJ+ B2 | TaxiBJ+ B3 | KTH B1 | KTH B2 | KTH B3 | RainNet B1 | RainNet B2 | RainNet B3 | PD B1 | PD B2 | PD B3 |
> > | ----------------- | ---------- | ---------- | ---------- | ------ | ------ | ------ | ---------- | ---------- | ---------- | :---: | ----- | ----- |
> > | ViT+SimVP         | 4.28       | 4.14       | 3.01       | 48.64  | 47.15  | 39.25  | 2.01       | 2.03       | 1.25       | 2.76  | 2.72  | 1.85  |
> > | SwinT+SimVP       | 4.25       | 4.09       | 2.94       | 46.02  | 45.62  | 37.16  | 1.95       | 1.91       | 1.19       | 2.71  | 2.69  | 1.76  |
> > | ViT+PredRNN       | 5.53       | 5.45       | 3.86       | 59.06  | 59.01  | 46.31  | 3.79       | 3.81       | 2.62       | 5.01  | 4.99  | 4.26  |
> > | SwinT+PredRNN     | 5.49       | 5.41       | 3.72       | 58.45  | 58.62  | 44.57  | 3.71       | 3.69       | 2.54       | 4.96  | 4.91  | 4.19  |
> >
> > We also conduct ablations from temporal augmentation and spatial augmentation two perspective to address your question more thoroughly:
> >
> > In our analysis, we implement three distinct ablation methods. For the first strategy, **A1**, we mix up only the causal patches at the current time point, not considering historical patches. For the second strategy, **A2**, we do not account for the time decay of historical patches, instead giving equal weight to all historical data (assigning a weight of 1 to each time point). As for the third strategy, **A3**, we focus on the historical patches of the region itself without considering the spatial context and causal patches, adding them with decay. We organize these ablation strategies as **A1**, **A2**, and **A3** respectively, and present the results (MAE) in a structured table format to facilitate a clear understanding of their impacts.
> >
> > | Model Combination | TaxiBJ+ A1 | TaxiBJ+ A2 | TaxiBJ+ A3 | KTH A1 | KTH A2 | KTH A3 | RainNet A1 | RainNet A2 | RainNet A3 | PD A1 | PD A2 | PD A3 |
> > | ----------------- | ---------- | ---------- | ---------- | ------ | ------ | ------ | ---------- | ---------- | ---------- | ----- | ----- | ----- |
> > | ViT + SimVP       | 2.89       | 2.72       | 3.02       | 37.64  | 35.25  | 43.26  | 1.16       | 1.08       | 1.25       | 40.86 | 35.64 | 50.62 |
> > | SWin + SimVP      | 2.72       | 2.61       | 2.94       | 36.56  | 34.12  | 42.64  | 1.05       | 0.99       | 1.16       | 39.34 | 34.16 | 49.13 |
> > | ViT + PredRNN     | 3.81       | 3.54       | 4.28       | 45.42  | 42.06  | 51.25  | 2.55       | 2.49       | 2.62       | 84.36 | 79.92 | 92.86 |
> > | SWin + PredRNN    | 3.72       | 3.41       | 4.15       | 44.32  | 41.17  | 50.14  | 2.45       | 2.48       | 2.53       | 82.32 | 77.43 | 90.95 |

---

> > > ### Comment · Reviewer_PkBe · 2023-11-22
> > >
> > > Sorry for the late reply. Thank you for the detailed response from the author; the major concern has already been addressed. While I acknowledge the contribution of this article, I encourage the author to include a comparison with TAU [1] in future experiments. I regard TAU as a stronger version of SimVP. In light of this, I have raised my score to 6.
> > >
> > > [1] Tan, Cheng, Zhangyang Gao, Lirong Wu, Yongjie Xu, Jun Xia, Siyuan Li, and Stan Z. Li. "Temporal attention unit: Towards efficient spatiotemporal predictive learning." In Proceedings of the IEEE/CVF Conference on Computer Vision and Pattern Recognition, pp. 18770-18782. 2023.

---

> > > > ### Author Response · Authors · 2023-11-22
> > > > **Response to Reviewer PkBe**
> > > >
> > > > Thank you for your valuable feedback and suggestion regarding our manuscript. In response to your comments, we have made the following revisions and clarifications:
> > > >
> > > > 1. We have included a reference to TAU in our newly submitted version to better highlight its significant contributions to the field.
> > > >
> > > > 2. In future versions of our work, we plan to incorporate experiments comparing our approach, Nuwa, with TAU to more effectively showcase the synergy between Nuwa and other significant works in the field."

---

> ### Author Response · Authors · 2023-11-14
> **Response to Reviewer PkBe _2**
>
> #### W3: Except for the KTH dataset, the performance gains are relatively low.
>
> When dealing with various datasets, we adhered to the original settings presented in their respective literature and adopted the appropriate data processing strategies. Firstly, for the TaxiBJ+ dataset, despite being introduced and utilized by us for the first time, we found that normalized data yielded more stable results in our experiments. Hence, we normalized this dataset before training. For the KTH dataset, given its relatively simple characteristics as grayscale images, we followed the design strategy from [4] and did not normalize the data, allowing for a wider range of numerical variation. Additionally, for the SEVIR dataset, we conformed to the methodology of [5], did not perform normalization, and chose data from the years 2016 to 2018 for our training. For the larger-scale RainNet dataset, we implemented the design strategy from [6-7] and carried out normalization. Lastly, for the univariate physical datasets PD and FireSys, as these data were recorded in video format [8], we did not normalize but opted to proceed directly with training. We posit that the KTH dataset, **with its grayscale imagery characteristics, exhibits a broad pixel range**. Furthermore, due to the dataset's relatively simple feature information, there is a significant improvement in performance.
>
> To help better understand Nuwa's capabilities, we select a 768x768 resolution cylinder dataset [9] to systematically validate the effectiveness of Nuwa. This dataset choice specifically addresses the performance of Nuwa in handling high-resolution scenarios, allowing us to demonstrate its robustness and accuracy in more demanding contexts.
>
> | Backbone    | MSE Ori | MSE +NuWa | MAE Ori | MAE +NuWa |
> | -- | -- | -- | --- | -- |
> | ViT         | 0.67    | 0.37 | 0.56    | 0.37 |
> | SwinT       | 0.61    | 0.34   | 0.55    | 0.33|
> | Rainformer  | 0.55    | 0.49  | 0.36    | 0.40 |
> | Earthformer | 0.49    | 0.33  | 0.43    | 0.32|
> | ConvLSTM    | 0.61    | 0.48  | 0.53    | 0.38 |
> | PredRNN-V2  | 0.70    | 0.61 | 0.49    | 0.39 |
> | E3D-LSTM    | 0.54    | 0.31  | 0.43    | 0.31 |
> | SimVP       | 0.40    | 0.31  | 0.37    | 0.30  |
>
> [4] Recognizing human actions: a local SVM approach.
>
> [5] Implicit Stacked Autoregressive Model for Video Prediction.
>
> [6] RainNet v1. 0: a convolutional neural network for radar-based precipitation nowcasting.
>
> [7]RainNet: A Large-Scale Imagery Dataset and Benchmark for Spatial Precipitation Downscaling.
>
> [8] Automated discovery of fundamental variables hidden in experimental data.
>
> [9] Lagrangian and Eulerian dataset of the wake downstream of a smooth cylinder at a Reynolds number equal to 3900.
>
> #### Q1: What are the limitations of current attention map methods for identifying causal regions?
>
> Existing methods for identifying the significance of tokens within vision transformer attention maps are all concentrated on image classification tasks. These methods [10-13] seek to ascertain the informativeness of a token by using the classification token [Cls] as a reference point. Transferring these approaches to spatio-temporal prediction tasks presents two serious issues: (1) Spatio-temporal prediction tasks, unlike classification tasks, do not require a [Cls] token. Furthermore, the final predictive information of classification task is aggregated in the [Cls] token, resulting in a lack of understanding of the individual importance of each token. (2) After summing the attention maps for each row, the total equates to one, making it challenging to identify important patches through traditional means of row averaging or summation.
>
> In our method, we select column summation to concentrate on the aggregate attention that sub-regions direct toward a particular patch. This technique is exceedingly straightforward and universally applicable. We design and integrate it into an upstream self-supervised reconstruction task, and then we incorporate it into a causal framework. To the best of our knowledge, our method innovatively introduces attention maps to establish a causal theory within the spatio-temporal observational image data framework, marking a pioneering contribution to the field.
>
> [10] Yuan, Li, et al. "Tokens-to-token vit: Training vision transformers from scratch on imagenet." *Proceedings of the IEEE/CVF international conference on computer vision*. 2021.
>
> [11] Pan, Bowen, et al. "IA-RED $^ 2$: Interpretability-Aware Redundancy Reduction for Vision Transformers." *Advances in Neural Information Processing Systems* 34 (2021): 24898-24911.
>
> [12] Rao, Yongming, et al. "Dynamicvit: Efficient vision transformers with dynamic token sparsification." *Advances in neural information processing systems* 34 (2021): 13937-13949.
>
> [13] Xu, Yifan, et al. "Evo-vit: Slow-fast token evolution for dynamic vision transformer." *Proceedings of the AAAI Conference on Artificial Intelligence*. Vol. 36. No. 3. 2022.

---

### Author Response · Authors · 2023-11-15
**Gratitude for Constructive Reviews and Hope for Enhanced Understanding**

We commerce by thanking the four reviewers for their thoughtful and constructive comments. We are really encouraged to see that the reviewers appreciate some positive aspects of our paper, such as an interesting problem to address (Reviewers PkBe, oBVm, TYFb) with strong motivation (Reviewer PkBe, oBVm, TYFb and zp2y), good technical novelty (Reviewer PkBe, oBVm, TYFb and zp2y), and promising/effective methodology (Reviewer oBVm, TYFb and zp2y). We also extend our gratitude to all reviewers for acknowledging the excellence and clarity of our writing. Your expertise significantly helps us strengthen our manuscript – this might be the most helpful review we received in years! In the following parts, we endeavor to provide individual responses to each reviewer. We sincerely hope that the additional experiments, model descriptions, and visualizations included can assist everyone in gaining a better understanding of our paper!

---

### Author Response · Authors · 2023-11-20
**Facilitating Reviewer-Author Communication**

We extend our heartfelt gratitude to the Area Chair (AC) and the reviewers for their rigorous efforts and commitment.  It is heartening to observe the recognition of our paper's strengths, including its clarity of writing, robust motivation, and technical innovation, as noted by Reviewers PkBe, oBVm, TYFb, and zp2y.

In response to the insightful comments received, we have diligently revised our paper, addressing the main concerns raised.  We are grateful for the positive reception and enhanced evaluations from Reviewer TYFb.  With great respect, we eagerly anticipate further constructive critiques from the reviewers to elevate our manuscript's caliber.

---

### Meta-Review · Area_Chair_ydTF · 2023-12-08

**Metareview:**

This paper introduces NuwaDynamics, a novel causal framework for spatio-temporal prediction tasks, leveraging self-supervised reconstruction tasks and interventions on non-causal regions. The paper is well-organized, with clear motivation and innovative contributions to the integration of causality concepts into spatio-temporal modeling. Visualizations demonstrate improved performance, especially in extreme weather perception. However, concerns about computational complexity, lack of ablation studies, and modest performance gains in some cases raise questions about the framework's broader impact. The authors have provided a thorough response to address the concerns.

All reviewers unanimously recommend accept.

Overall, it is a promising contribution to the ICLR community.

**Justification For Why Not Higher Score:**

AC thinks that it's better to tone down that it tackles the "causality" of spatial dynamics.

**Justification For Why Not Lower Score:**

Interesting and high-quality paper.  All the reviewers recommends accept.

---

### Decision · Program_Chairs · 2024-01-16

Accept (spotlight)